# High-precision laser spectrometer for multiple greenhouse gas analysis in 1 mL air from ice core samples

Bernhard Bereiter[1,3], Béla Tuzson[1], Philipp Scheidegger[1,2], André Kupferschmid[2], Herbert Looser[1], Lars Mächler[3], Daniel Baggenstos[3], Jochen Schmitt[3], Hubertus Fischer[3], and Lukas Emmenegger[1]

[1]Laboratory for Air Pollution / Environmental Technology, Empa - Swiss Federal Laboratory for Materials Science and Technology, 8600 Dübendorf, Switzerland
[2]Transport at Nanoscale Interfaces, Empa - Swiss Federal Laboratory for Materials Science and Technology, 8600 Dübendorf, Switzerland
[3]Climate and Environmental Physics and Oeschger Center for Climate Research, University of Bern, 3012 Bern, Switzerland

**Correspondence:** Béla Tuzson (bela.tuzson@empa.ch)

**Abstract.** The record of past greenhouse gas composition from ice cores is crucial for our understanding of global climate change. Future ice core projects will aim to extend both the temporal coverage (extending the time scale to $1.5\,\mathrm{Myr}$) and the temporal resolution of existing records. This implies a strongly limited sample availability, increasing demands on analytical accuracy and precision, and the need to reuse air samples extracted from ice cores for multiple gas analyses. To meet these requirements, we designed and developed a new analytical system that combines direct absorption laser spectroscopy in the mid-infrared with a quantitative sublimation extraction method. Here, we focus on the high-precision dual-laser spectrometer, for the simultaneous measurement of $CH_4$, $N_2O$ and $CO_2$ concentrations, as well as $\delta^{13}C(CO_2)$. Flow-through experiments at $5\,\mathrm{mbar}$ gas pressure demonstrate an analytical precision $(1\,\sigma)$ of $0.006\,\mathrm{ppm}$ for $CO_2$, $0.02\,‰$ for $\delta^{13}C(CO_2)$, $0.4\,\mathrm{ppb}$ for $CH_4$ and $0.1\,\mathrm{ppb}$ for $N_2O$, obtained after an integration time of $100\,\mathrm{s}$. Sample-standard repeatabilities $(1\,\sigma)$ of discrete samples of $1\,\mathrm{ml}$ STP amount to $0.03\,\mathrm{ppm}$, $2.2\,\mathrm{ppb}$, $1\,\mathrm{ppb}$ and $0.04\,‰$ for $CO_2$, $CH_4$, $N_2O$ and for $\delta^{13}C(CO_2)$, respectively. The key elements to achieve this performance are a custom-developed multipass absorption cell, custom-made high-performance data acquisition and laser driving electronics, and a robust calibration approach involving multiple reference gases. The assessment of the spectrometer capabilities on repeated measurement cycles of discrete air samples – mimicking the procedure for external samples such as air samples from ice cores – was found to fully meet our performance criteria for future ice core analysis. Finally, this non-consumptive method allows the reuse of the precious gas samples for further analysis, which creates new opportunities in ice core science.

## 1 Introduction

Precisely monitoring the current anthropogenic rise of the greenhouse gas ($CO_2$, $CH_4$ and $N_2O$) concentrations is essential for the implementation of the Kyoto (UNFCCC, 1998) and Paris (UNFCCC, 2015) agreements. Therefore, several monitoring networks have been established, comprising both ground- and satellite-based instrumentation to accurately measure the greenhouse gases with high temporal and spatial resolution. For ground-based monitoring, where essentially unlimited volums of

sample are available, continuous infrared spectroscopic analysis is gaining increasing importance, as it allows for high time resolution with a minimum of sample preparation and multiple greenhouse gases in a single instrument (e.g. McManus et al., 2010; Hammer et al., 2013; Hundt et al., 2018). In recent years, these techniques have been further developed, also enabling isotopic measurements of $CO_2$, $CH_4$ and $N_2O$ (e.g. Tuzson et al., 2011; Prokhorov et al., 2019; Eyer et al.; Ibraim et al., 2017).

Today's changes are precisely monitored with constantly improving global measurement networks, providing strong constraints on anthropogenic and natural emissions. However, such direct atmospheric observations only began in the late 1950s (Graven et al., 2013), and therefore, other sources for data, such as temperature and greenhouse gases, are needed to validate and constrain climate models when covering many centuries or even millennia. Moreover, processes controlling the natural range and variability of greenhouse gases can only be fully assessed when the greenhouse gas record is extended over the full range of climate variations, representative of long-term centennial, millennial, orbital up to weathering time scales. An extension of the observation record, spanning the last $800\,kyr$, has become possible using polar ice cores, from which small samples of past atmospheric air can be extracted. Thus, a reconstruction of the $CO_2$, $CH_4$ and $N_2O$ records over the entire anthropogenic era (Rubino et al., 2013; MacFarling Meure et al., 2006) and further back in time over past glacial-interglacial cycles covering up to the last $800\,kyr$ was realized (Loulergue et al., 2008; Schilt, 2013; Bereiter et al., 2015; Petit et al., 1999; Lüthi et al., 2008). Apart from greenhouse gas concentrations in ice cores, which are often determined by GC techniques, the precise quantification of their isotopic composition using mass spectrometric (MS) or coupled gas chromatography - mass spectrometric (GC-MS) analyses of ice core samples has become possible in recent years (Schmitt et al., 2012; Schilt et al., 2014; Bock et al., 2017; Bauska et al., 2018). However, these methods involve tedious and time-consuming separation of individual gas species from the air matrix. As an alternative approach, the application of a mid-IR tunable diode laser spectrometer for discrete $CO_2$ analyses on ice cores had been pioneered at the University of Bern in the 1970s (Lehmann et al., 1977; Neftel et al., 1982). This technique avoids the separation of $CO_2$ from the gas matrix and enables concentration measurements on small ($<10\,g$) of ice samples (Bereiter et al., 2015; Güllük et al., 1997, and references therein). Recently, cavity-ring-down laser spectroscopy in the near-IR was applied for continuous online measurements of $CH_4$ in ice cores (Chappellaz et al., 2013; Rhodes et al., 2015). However, as these analyses involve a gas separation step from a continuous water stream, solubility effects require external calibration of such melt water based online measurements.

In 2019, the European "Beyond EPICA Oldest Ice Core" project was started by partners from ten European nations with the goal to retrieve a continuous Antarctic ice core going back over the last $1.5\,Myr$ (Fischer et al., 2013). For the first time, this ice core will allow to reconstruct the atmospheric changes that occurred over the so-called Mid Pleistocene Transition (approximately $1.2 - 0.9\,Myr$ ago), when the cyclicity of glacial/interglacial cycles changed from $40$ to about $100\,kyr$, and the amplitude of continental glaciation during ice ages substantially increased. Due to glacier flow, the ice at the bottom of the ice sheet, where the very old ice is preserved, experienced extreme thinning, i.e. about $15\,kyr$ of climate history are compressed into only $1\,m$ of ice core (Fischer et al., 2013) thus implying a limited availability of sample for greenhouse gas and other analyses. Similar extreme thinning is observed in the bottom-most sections of high accumulation coastal Antarctic ice cores, which allow for temporal extension of high-resolution greenhouse gas records. Accordingly, such extraordinary glaciological conditions require novel analytical approaches to maximize temporal resolution of the records, while minimizing

sample consumption and, at the same time, making no compromises on precision and repeatability of the gas analyses. Thus, the various greenhouse gas analyses, previously done on several pieces of ice, have to be combined and the sensitivity of the analytical methods improved, and whenever possible reuse of the gas after the analysis for other analytical purposes should be pursued.

These goals motivated our development of a novel approach that combines the high-resolution and selectivity of a laser absorption spectrometer with the quantitative and continuous extraction of the air enclosed in ice cores by a unique sublimation technique. In this publication, we will present in detail the laser absorption spectrometer and its performance, while the continuous sublimation extraction system providing a cm-scale vertical resolution will be described in a separate paper (Mächler, 2020).

In atmospheric sciences, laser spectroscopy is a well-established analytical method for high-precision trace gas concentration and isotope ratio measurements. However, the stringent requirements related to ice core analysis still remain highly challenging. In the present study, our aim was to establish an analytical tool, which enables the simultaneous quantification of $CO_2$, $CH_4$ and $N_2O$ concentrations as well as the stable carbon isotopic composition of $CO_2$ in ice core-derived air sample of only $1$–$2\,mL$ STP without separation of these gases from the air matrix. In order to allow for authoritative interpretation of the observed glacial/interglacial changes in the biogeochemical cycles of these three greenhouse gases, a signal-to-noise ratio of better than 5 for the centennial to multi-millennial variations found in ice cores for $CO_2$, $CH_4$, $N_2O$, and $\delta^{13}C(CO_2)$ over the last glacial cycles is required. This results in precision targets of these parameters for high-quality ice core analyses of $0.5\,ppm$, $2\,ppb$, $2\,ppb$, and $0.04\,\permil$, respectively. These targets are either comparable or better than the best ice core analysis systems available to date. Moreover, the option of cryogenically recollecting the air sample after the laser spectroscopic analysis was also foreseen.

In the following, we present the technical details of the developed instrument including the optical design, the custom-made absorption cell and electronics, the instrument periphery as well as the custom-made standard gases needed for the calibration. Finally, a detailed characterization and calibration for $1\,mL$ STP air samples is given, demonstrating the excellent analytical capabilities of the spectrometer, and including steps towards a calibration scheme that can handle the variable sample volumes and pressures that are expected from ice core samples. We stress that the sample size, gas handling and sample introduction as well as the calibration scheme presented here have been specifically designed and optimized for the use on discrete ice core air samples of 1-2 mL STP that will be provided by the sublimation extraction system that is being developed in parallel.

## 2  Methods

The targeted multi-species capabilities of our instrument is achieved by using a dual-laser concept (see e.g. McManus et al., 2011, 2015), where two distributed feedback quantum cascade lasers (DFB-QCLs) with distinct frequencies are combined and coupled into a multipass absorption cell (MPC). The lasers were selected to fulfill fundamental line selection criteria (see e.g. Tuzson et al., 2008) regarding spectral interference issues, limitations in laser tuning capabilities ($<4\,cm^{-1}$), and achievable signal-to-noise ratio (SNR) under the targeted sample conditions. Furthermore, the line selection (see Table1) considers the fact

**Table 1.** Spectroscopic parameters of the absorption lines selected for this work. The molecule $ID$ (abbreviated code for isotopologues), line positions $\nu$ (cm$^{-1}$) as well as the lower state energies $E''$ (cm$^{-1}$), line strengths $S$ ($10^{-20}$ cm$^{-1}$/(cm$^2$molecule$^{-1}$)), and Einstein $A$-coefficients (s$^{-1}$) are from the HITRAN 2016 database (Gordon et al., 2017).

| Species | $ID$ | $\nu$ | $S$ | $E''$ | $A$ |
|---|---|---|---|---|---|
| $^{12}CO_2$ | 21 | 2301.680904 | 2.71 | 1276.4476 | 202.3 |
| $^{13}CO_2$ | 22 | 2302.308939 | 2.62 | 273.8809 | 187.8 |
| $N_2O$ | 41 | 1301.684840 | 15.40 | 175.9536 | 6.1 |
| $CH_4$ | 61 | 1302.044313 | 6.45 | 219.9199 | 2.2 |

that $CH_4$ and $N_2O$ share a region of relatively strong absorption lines around 7.7 μm, which gives access to both species within the spectral coverage of a DFB-QCL. For $CO_2$ (including the two most abundant isotopologues), we use the region around 4.3 μm, with a special focus on absorption lines that have comparable intensities. In this spectral region, there are mainly two options for $CO_2$ that fulfill our selection criteria: The lines proposed by Tuzson et al. (2008) around $2310\,\mathrm{cm}^{-1}$ and the ones around $2302\,\mathrm{cm}^{-1}$ that we use in this work. The latter were chosen because of their higher line intensity which helps

reaching the required SNR despite the small sample size. However, it should be recognized that comparable absorption for the $CO_2$ isotopologues means that their ro-vibrational transitions have rather different ground state energies (see also Table1). Therefore, the spectroscopically retrieved $\delta^{13}$C-values exhibits a larger temperature sensitivity (Tuzson et al., 2008). In our case, we estimate a temperature sensitivity of the isotope ratio of about $16\,‰\,\mathrm{K}^{-1}$. Although it may be difficult to maintain long-term stability of the absorption cell to better than 0.01 K, changes in the cell temperature can be measured with a precision

on the order of mK using thermistors. Since the measured temperature is used continuously for interpreting the absorbance spectra, the temperature dependence of the line strength is not a major impediment to obtain isotope ratio precision better than $0.02\,‰$ for $\delta^{13}CO_2$.

## 2.1 Optical design of the QCLAS

The concept and the final layout of our optical setup are shown in Fig. 1. During the development, first a 3D CAD model cou-

105 pled to ray-tracing simulations was used that allowed easy testing of different design options and optimizing beam propagation within the available space. Key factors were the beam-shaping and efficient coupling of both laser beams into the same absorption cell, which is a custom-made astigmatic Herriott multipass cell (see Sect. 2.1.1). The QCLs (Alpes Lasers, Switzerland) were packaged in high-heat-load (HHL) housings with embedded thermoelectric cooler (TEC) and collimating lens, and the output beam properties (size and profile) were characterized using a mid-IR beam profiler (WinCamD-IR-BB, DataRay, USA).

The empirical values were then used to define the light sources in the ray-tracing software (FRED, Photon Engineering, USA). This allowed a realistic beam propagation along the optical path and, thus, led to a reliable selection of the steering and shaping elements, which best fulfill the entrance conditions of the MPC (see e.g. McManus, 2007). The infrared beam is focused near the center of the cell and it's beam waist (374 μm at 4.34 μm wavelength) is closely matched to the nearly confocal cavity, such

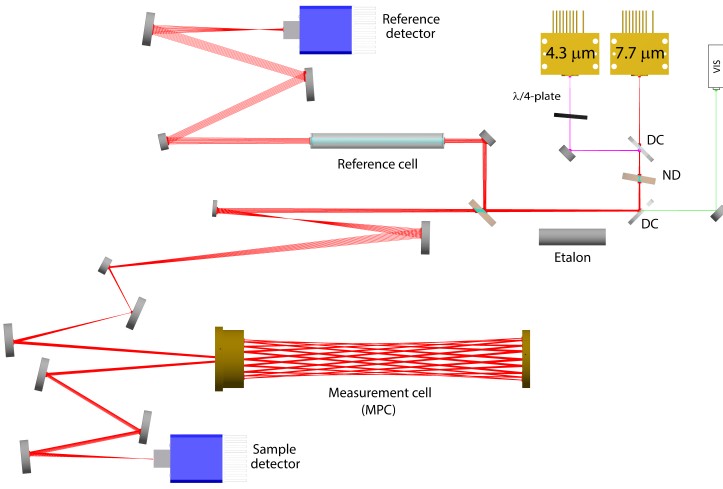

**Figure 1.** Optical layout of the dual-QCL system showing all relevant optical elements (where DC refers to dichroic mirror, ND to neutral density filter) and the optical path (colored lines) of the laser beams as simulated by the ray-tracing software (FRED, Photon Engineering).

that the reflected beam maintains a nearly constant size at the mirrors while propagating in the cell. The coupling of the 7.7 $\mu$m wavelength is slightly off from the ideal case and the realized beam-waist of 1.1 mm is about twice as large as theoretically expected. The obtained fringe level (see residual plot in Fig. 2) is, nevertheless, very low, indicating that the MPC is rather tolerant to such mismatch.

In a first section, a beam of three lasers is generated by using a custom-made dichroic mirror (LohnStar Optics, USA) to efficiently combine the two mid-IR beams first, followed by a second dichroic mirror (Quantum Design, Switzerland) to couple in a red trace laser, which is used for alignment purposes. After the beam combination, the beam is split into two paths: a 'sample' path that goes through the MPC, and a 'reference' path, which is directed through a reference cell filled with a predefined gas mixture. The splitting ratio is such that the extra losses due to the multiple reflections within the MPC are compensated for, and both the sample and reference detectors (PVM-2TE-8-1x1-TO8-wZnSeAR-70, Vigo Systems, Poland) receive approximately the same optical power. The optical elements downstream of the beam splitting are for beam shaping and steering. The reference cell is mounted on a remotely controllable flip-mount, and it is taken out of the beam once per measurement cycle to obtain the laser emission profile used as signal background for the reference path.

Investigations of the laser emission properties regarding intensity and frequency stability revealed an increased sensitivity of the QCLs to optical feedback when they are driven with currents near their lasing threshold (104 mA). This was more pronounced for the QCL emitting at 4.3 $\mu$m and led to marked instabilities of the emission frequency (jitter). Therefore, the QCLs are driven at high current (160 mA), and a quarter wave plate is placed in the front of the 4.3 $\mu$m QCL to further reduce optical feedback. High-current operation of QCLs implies increased optical output power, which is usually beneficial in terms of SNR in the mid-IR. However, in our case, the QCLs provide so much optical power (54 mW) that (i) the linear range of the

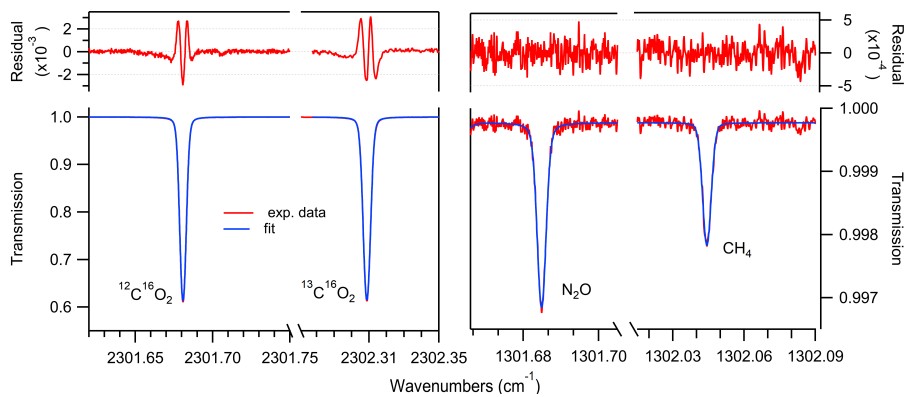

**Figure 2.** Example of measured transmission spectra (red), Voigt-profile fit (blue) and residual (top) for the scanned ranges at $4.3\,\mu m$ (left) and $7.7\,\mu m$ (right) for a gas at $5\,mbar$ with concentrations representative of ice core samples, i.e. using standard gas #6 (see Table 2).

detector is exceeded, and (ii) the absorption process in the low-pressure sample gases becomes saturated. To avoid these effects, a custom-made wedged neutral density filter (ND = 1.0) is placed between the two dichroic mirros. This way, the optical power of both QCLs is reduced by $90\,\%$ before entering into the MPC, while the transmitted optical power reaching the detector is $0.6\,mW$. As we will show later, this laser intensity was still not low enough to completely eliminate the saturation effect, but we refrained from further laser intensity reduction, because this would have a negative impact of the SNR value, jeopardizing the targeted precision. However it is important to note that we operationally correct for the small saturation effects within our calibration scheme (see Sect. 3.3).

To decouple the optical setup from external temperature changes, the optical breadboard is water-cooled and maintained at $20 \pm 0.005\,°C$ using a close-loop thermochiller (Oasis Three, Solid State Cooling Systems, USA). Furthermore, the whole setup is enclosed in a sealed and thermally insulated case (Fibox, Switzerland), which is purged with $CO_2$-free air (resulting in a stable residual background of about $15\,ppm\ CO_2$) to minimize light absorption outside the sample and reference cells. These factors are significantly improving the long-term stability of the measurements.

### 2.1.1 Custom-made multipass absorption cell

The absorption cell is a key component of any absorption laser spectrometer. It not only defines the supported optical path length (OPL), but also the volume in which the light can interact with the molecules of the sample gas. As the SNR scales with the increase of OPL, extended paths are usually achieved by using beam folding concepts, such as multipass cells (MPCs). Beam folding, however, is also an important source of optical noise in the system and needs careful consideration. In our application, the precision targets of about $2\,ppb$ for $CH_4$ and $N_2O$ at ambient concentration and low pressure ($<10\,mbar$) can be achieved with an estimated minimal optical path length of around $30\,m$. We considered these parameters as criteria for the required OPL, because of their significantly lower absorption signal compared to that of the $CO_2$ isotopologues (see Fig. 2 and note the different scaling in the transmission axis). These requirements are largely fulfilled by a commercially

available solution (AMAC-36, Aerodyne Research, USA), which is currently the state-of-the-art MPC for high-precision trace gas measurements (McManus et al., 2010) and which was our first choice for the prototype instrument. However, the very limited sample volume and the aim of cryogenic re-collection of the sample after measurement create additional demands on the cell: i) it must be very leak-tight, ii) the cell inner surface must be highly inert, and iii) the cell should have a minimum dead volume, i.e. volumes outside the optically active region. Since the AMAC-36 is primarily aimed at atmospheric monitoring applications using flow-through mode, the above aspects were not fully met. This became evident when using the AMAC-36 in static mode where we observed a continuous decrease of $0.2\,\mathrm{ppm\,min^{-1}}$ in the $CO_2$ concentration. Furthermore, despite the leak-tightness ($1 \times 10^{-6}\,\mathrm{mbar\,L\,s^{-1}}$) of the MPC, the evacuation of the cell took significantly longer compared to a simple high-vacuum-proofed volume of similar size. The response time was strongly dependent on the conditioning history of the cell, but even by baking, purging, and evacuating the cell over few days, the pump-down time of the cell after filling with a sample gas took at least 20 minutes. Hence, we concluded that these characteristics were caused by adsorption processes on the cell inner surfaces and by internally closed volumes acting as "gas buffers". Trying to fix these issues by coating the surfaces and modifying the constructional design of this cell appeared to be very challenging. Therefore, we decided for the development of a custom-made cell, taking into consideration all of the above requirements in its design.

The basic design difference of our cell with respect to the AMAC-36 is the cell-body milled from solid stainless steel, which allows to shape the cell inner surface as a cuboid that matches the envelope of the optical pattern of the laser beam between the two astigmatic mirrors. This leads to a $30\,\%$ reduction of the inner volume (to $166 \pm 5\,\mathrm{cm^3}$) compared to a simple cylindrical geometry and thus increases the pressure of the air sample, which is crucial to reach the targeted precision. A further significant benefit of the single-piece cell body is the possibility to manufacture most of the mechanics, required for mirror mounting and aligning, out of the same piece. This allows minimizing manufacturing tolerances and reducing the degrees of freedom for mirror mounting to axial shift and rotation of the back mirror. Thus, this concept allows for a simpler alignment mechanism, and hence it also minimizes the dead volume behind the rear-mirror that would be required otherwise.

Beside the cell-body, the astigmatic mirrors were also reconsidered. The calculation of the geometry of the astigmatic mirror pair is based on the paraxial matrix approach (McManus et al., 1995; McManus, 2007). In general, for a given mirror geometry, there is a large number of possible reflection patterns supported for varying mirror distances and tilt angles. For the search of suitable mirror geometries, we set the following boundary conditions: i) a low volume for the targeted $30\,\mathrm{m}$ OPL, and ii) a reflection pattern that generates a minimum of optical fringes, or fringes with frequencies that substantially differ from the absorption line widths (about $100\,\mathrm{MHz}$). The first condition is well met by a base length of $20\,\mathrm{cm}$ and mirror diameters of $4\,\mathrm{cm}$. For the second aspect, we first generated a pattern map (McManus et al., 2011) with all possible combinations of mirror curvatures and tilt angle, then calculated for individual patterns the spatial separation of each reflection spot with respect to its neighbors, and searched for candidates with a high separation level. The expected interference frequencies were derived based on the optical path difference of the neighboring spots. Apart from the lowest possible overlap of the individual reflection spots, we also took into account the optical path difference of the neighboring spots and avoided those patterns where the interference fringe frequency generated by the optical path difference is comparable to the width of the absorption line. Overlaps of the beam spots with small pass number (e.g. 4 and 6) differences are especially problematic. Searching for patterns that fulfil

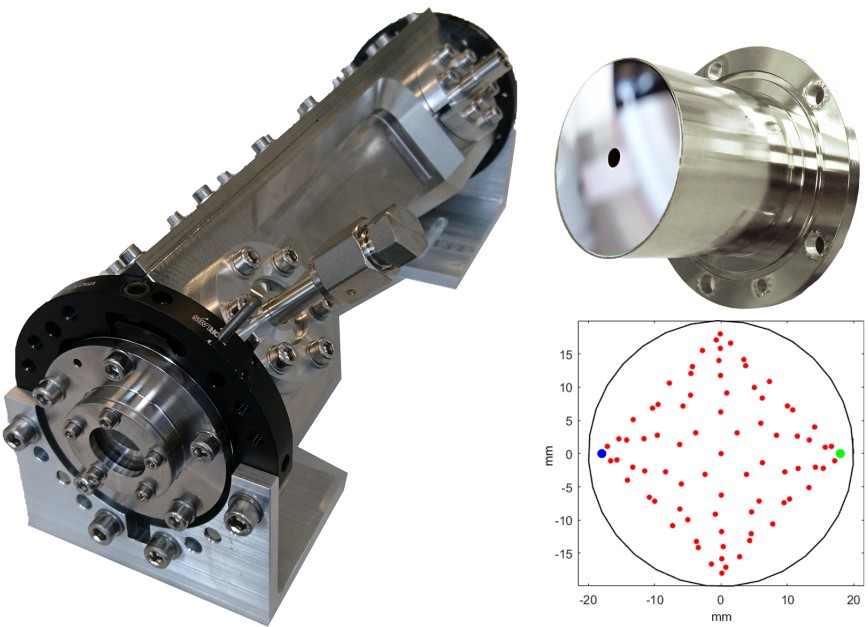

**Figure 3.** The low-volume and high-vacuum-sealed optical MPC developed and manufactured for the spectrometer. Left: Photograph of the multipass absorption cell with its entry window and the gas inlet in the foreground. The cell inner body along the optical axis is shaped as a cuboid that matches the envelope of the optical pattern of the laser beam between the two astigmatic mirrors. This design approach leads to a 30 % reduction of the inner volume compared to a simple cylindrical/rectangular case. Right-top: Photograph of the custom-made astigmatic mirror with the entry-hole in the center. The mirror body design allows for a direct and leak-tight attachment to the cell thereby minimizing the dead volume. Right-bottom: Reflection pattern (red dots) on the two mirrors as used in the current configuration. The blue and green dots represent the first and last reflection, respectively.

these criteria are expected to result in reduced optical fringe level and have less impact on the absorption line retrieval. In consideration of all these aspects, we selected the pattern shown in Fig. 3, which is characterized by 162 reflections. The mirrors were manufactured by diamond turning and post polishing a NiP coated aluminum substrate (LT Ultra-Precision Technology, Germany). Finally, a broadband high-reflectivity coating (Pleiger Laseroptik, Germany) was applied to achieve approx. 99 % reflectivity for both selected IR-spectral ranges and about 98 % for the visible range. The exact optical path length ($34.134 \pm 0.003\,\mathrm{m}$) was measured by coupling in a commercial laser distance meter (Disto D510, Leica Geosystems, Switzerland). The high-vacuum-proofed construction assures leak-rates below $1 \times 10^{-7}\,\mathrm{mbar\,L\,s^{-1}}$ and fast pump down times (from $5\,\mathrm{mbar}$ to $0.01\,\mathrm{mbar}$ in $90\,\mathrm{s}$). This is a prerequisite for rapid switching between sample and standard which contributes to a high accuracy in applications with small, discrete samples (see Sect. 3) such as those derived from ice cores.

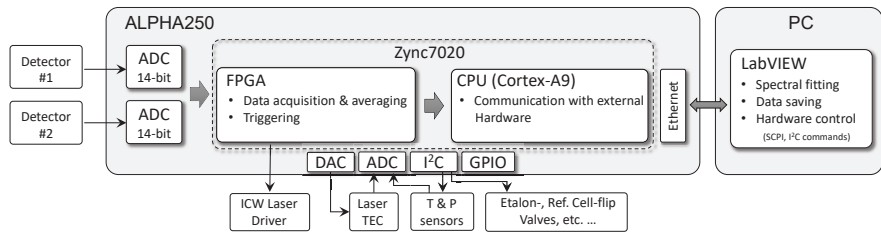

**Figure 4.** Schematic of the electronics setup. The core of the system is a system-on-chip (SoC) embedding a dual-core ARM processor and a FPGA, which triggers the laser driver, reads the ADCs and processes the detector signals in real-time. All processes are synchronized using the same clock generator. The hardware internal communication is based on the Inter-Integrated Circuit ($I^2C$) protocol. The CPU communicates with the host PC and drives other hardware components, which are less critical with respect to the timing.

## 2.2 Hardware design and data processing

Our laser driving and data processing electronics approach are described in detail elsewhere (Liu et al., 2018; Tuzson et al., 2020) and only a brief summary is given here. The two QCLs are driven in intermittent continuous wave (ICW) mode with time-division multiplexed timing (Fischer et al., 2014). By applying the current pulses, the laser emission frequency is rapidly tuned, allowing for complete spectral scans within $75\,\mu s$. The tuning-rate is transformed into a linearized frequency scale by using the transmission spectrum of a 2-inch solid Germanium etalon, which is mounted on a custom-made flip-mount. Repeated measurements indicate that the QCL tuning characteristics are highly stable over time and new characterizations are only needed after major interventions on the system.

Hardware control and data acquisition electronics deployed here are similar to those of Liu et al. (2018). The main difference is that here we use a more powerful programmable board (Alpha250, Koheron, France), built around a field-programmable gate array (FPGA), which features faster sampling rate ($250\,\mathrm{MSs}^{-1}$) and a higher bandwidth ($100\,\mathrm{MHz}$) analog front-end with dual-channel 14-bit ADCs and 16-bit DACs. This upgrade was necessary because of the required higher spectral resolution (narrow line widths due to low-pressure samples) and the dual-path configuration resulting in two simultaneous inputs to the ADC (see Fig. 4). The FPGA firmware (VHDL-Code) and the Linux service routines (C-Code) were custom developed. The FPGA contains a state-machine that is clocked eight times slower than the sample clock, which makes routing less critical, i.e. larger data path delays are tolerated. For higher flexibility in data acquisition, several user-defined time-windows within a spectral scan are supported. The summation of consecutive spectra is implemented with digital signal processors (DSP) and dual port block random access memory (BRAM). The spectra are then transferred from the programming logic via direct memory access (DMA) into the DDR-RAM of the processing unit. These data are then sent to an external computer via a TCP/IP-interface for post-processing.

For the spectral analysis, we used the averaged data of 5000 spectra corresponding to $1\,\mathrm{Hz}$ time resolution. In total, five acquisition windows were defined over the two laser pulses. One data acquisition window is used to record the detector signal while the lasers are not turned on yet. This signal is used to normalize the follow-up spectra that are collected in the other

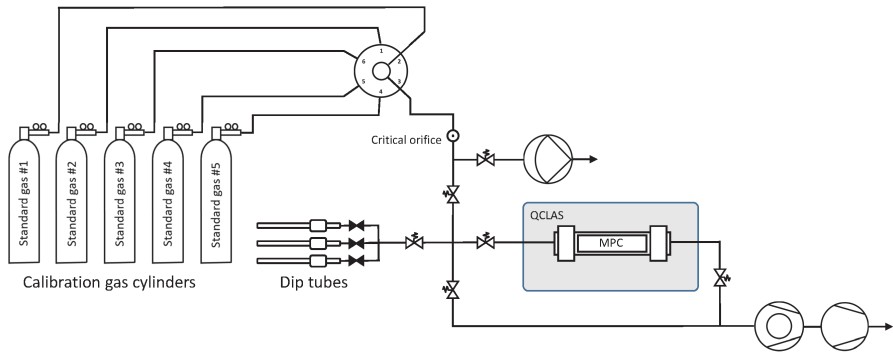

**Figure 5.** Gas handling system primarily used to characterize and calibrate the QCLAS instrument (schematically indicated by the gray rectangle), but also to introduce small air samples trapped in dip tubes that contain air extracted from ice core samples. The gas handling system is made out of ultra-high vacuum (UHV) stainless steel tubing and uses exclusively VCR- and KF-seals downstream of the critical orifice. The majority of the valves are pneumatically actuated valves (Fujikin Inc., Japan). For the flow-through measurements, a bypass with a metering valve (not shown) was used to reduce the flow.

four windows within the laser scan. The concentrations of the individual species are retrieved in real-time by fitting a Voigt function to the measured spectra using the Levenberg-Marquardt least-squares algorithm implemented in LabVIEW (National Instruments, USA). The spectral line intensity and the broadening parameters are taken from the HITRAN2016 database (Gordon et al., 2017), whereas the gas pressure and temperature are measured and their values are used in the fitting algorithm
to calculate the Doppler- and Lorentzian-widths and determine the number density of the target molecules. The corresponding transmission spectra with the associated fitted curves using Voigt profiles are shown in Fig. 2.

## 2.3    Gas handling hardware

A fully automated gas handling system has been built around the spectrometer as depicted in Fig. 5. The system is equipped with two pumps (turbo-molecular pump station and diaphragm pump) used to pull reference gases from the cylinders through
parts of the system at a steady flow, independently of the measurement cycle, or to evacuate the MPC and the manifold-line between individual discrete samples. A multi-port selector (6-port, Vici Valco Instruments Inc., Switzerland) with individual vents allows switching between the standard gas cylinders while maintaining an uninterrupted gas flow. After the multi-port selector, the standard gas is pulled through a flow restriction (critical orifice), where the pressure drops from about $2 \, \mathrm{bar}$ absolute to the $\mathrm{mbar}$-level. The critical orifice size ($20 \, \mu\mathrm{m}$) as well as the output pressure of the regulators were all carefully
selected to avoid isotopic fractionation as the gases flow through the system. A series of experiments was conducted in flow-through and static regime to investigate potential fractionation effects due to flow-restriction elements such as critical orifice and the optional metering valve used in the flow-through mode. Their effect was found to be lower than our detection limit. With this setup it is possible to operate the instrument both in flow through or in batch mode for discrete samples. In the latter, the evacuated MPC is filled to any target pressure between 2.0 and $10.5 \, \mathrm{mbar}$ with an uncertainty of $\pm 0.04 \, \mathrm{mbar}$. For this, a

pressure sensor (CMR 363, Pfeiffer Vacuum, Switzerland) monitors the sample pressure in the MPC at $10\,Hz$ and its output
signal is used to control the pneumatic valves and decouple the cell when the preset pressure is achieved.

## 2.4  Custom-made standard gases

Direct absorption laser spectroscopy establishes a well-defined relation between the unknown amount fraction and the measured
absorbance. This relation only contains directly measurable physical quantities (temperature, pressure, optical path length, line
amplitude) and molecular properties, and thus it makes the technique – in terms of metrology – a calibration-free method to
determine the amount fraction of a trace gas (e.g. Buchholz et al., 2014). However, as any "real" system, this approach is
also affected by for example detector non-linearity, drifts, optical fringes or general instrumental response that may change
over time. The relevance of these effects is gaining significance as the requirements regarding instrument performance become
more stringent, which is especially pronounced for measurements of stable isotope ratios. Relatively stable and predictable
instrument-related artifacts can be accounted for by a well-designed calibration scheme, which may also provide traceability
to SI-units or an established calibration scale. A widely-used approach is multi-dimensional bracketing with reference gases
of known isotopic composition. This targets the most critical dependencies and links the measured spectroscopic values to an
international scale in a range that covers the expected variations occurring in the samples. Relying on our previous experience
in high-accuracy, traceable, long-term, and robust calibration of laser spectrometers for isotope ratio measurements (see e.g.
Tuzson et al., 2011; Sturm et al., 2013), we adopted here the strategy of delta-based calibration. For this, we prepared a
set of calibration gases with specific mixtures. The aim was to realize a two-dimensional bracketing for two major goals: i)
reliably link the spectrally-derived isotope ratios to the internationally accepted Vienna PeeDee Belemnite (VPDB) scale, and
ii) account for the dependency of the retrieved $\delta^{13}C(CO_2)$ values on the $CO_2$ concentration (Griffith, 2018). In addition,
the calibration gases were designed to also provide bracketing of the other two trace gases ($N_2O$ and $CH_4$). As a result,
five standards were prepared (see Table 2), two pairs each having almost identical $\delta^{13}C(CO_2)$ (values fixed at $-3.7\,‰$ and
$-10\,‰$), but different $CO_2$ concentrations covering pre-industrial/glacial-interglacial atmospheric composition ($160\,ppm$ and
$350\,ppm$). The fifth standard gas was prepared in such a way that it falls in the middle of the corner values for both $\delta^{13}C(CO_2)$
and $CO_2$ concentration, as indicated in Fig. 6. Here, the concentrations of $N_2O$ and $CH_4$ are indicated by colors (red, green,
yellow) that qualitatively reflect the respective trace-gas content (high, medium, and low).
The production of standard gases involved two main steps: (i) gravimetric production of approximate mixtures, i.e. within
$10\,\%$ of the target, and (ii) subsequent quantification of these mixtures by established, traceable methods. For the first step, the
approximate mixtures were realized by sequentially freezing out the targeted pure greenhouse gases in cylinders, followed by
dilution with greenhouse gas-free atmospheric air. The procedure in detail was: i) evacuate a $50\,L$ aluminum gas cylinder, ii)
place the cylinder in a liquid nitrogen bath, iii) cryogenically collect pure $CO_2$ (with one or two different isotopic ratios), $CH_4$
and $N_2O$ from gas cylinders, iv) cryogenically collect matrix air using a modified zero-air generator that provided (almost)
greenhouse gas-free atmospheric air with unchanged $N_2/O_2/Ar$ ratio. This is an important aspect to avoid any potential bias
caused by uncertainties in pressure broadening effects of these species (e.g. Nara et al., 2012). One important advantage of this
cryogenic trapping is that it enables the production of large volumes ($6500\,L$ STP) of standard gases, because the gas cylinders

**Table 2.** List of custom-made standard gases produced for the calibration of the QCLAS. The concentrations and isotope ratio values cover the range of expected variations found in ancient air samples from ice cores. The standard #2 is only matrix air that was used to generate the various standard mixtures. The uncertainty of the values are given in parentheses.

| Cylinder # | $CO_2$ (ppm) | $CH_4$ (ppb) | $N_2O$ (ppb) | $\delta^{13}C(CO_2)$ (‰) |
|---|---|---|---|---|
| 1. | 248.803 (0.01) | 528.576 (0.05) | 241.987 (0.250) | -6.639 (0.037) |
| 2. | 0.201 (0.01) | 0.121 (0.04) | 172.692 (0.100) | n.a. |
| 3. | 157.709 (0.02) | 331.178 (0.04) | 189.190 (0.310) | -3.722 (0.015) |
| 4. | 167.044 (0.02) | 810.950 (0.03) | 340.387 (0.020) | -9.880 (0.017) |
| 5. | 345.514 (0.03) | 779.778 (0.14) | 325.678 (0.040) | -3.697 (0.023) |
| 6. | 239.213 (0.03) | 527.185 (0.08) | 236.351 (0.280) | -6.659 (0.020) |
| 7. | 341.820 (0.05) | 339.236 (0.07) | 167.381 (0.330) | -10.074 (0.014) |

can be filled up to their maximum filling pressure limit ($200\,\mathrm{bar}$). This permits the production of a set of cylinders with standard
gases that will last for many years. In a final step, the cylinders were analyzed against NOAA/ESRL standards at the World
Calibration Centre (WCC) at Empa for $CO_2$, $CH_4$ (CRDS, G1301, Picarro Inc., USA) and $N_2O$ (QC-TILDAS, Aerodyne
Research Inc., USA) concentrations several times over the following weeks to investigate potential drifts that may appear
due to the cryogenic filling. The results show that all cylinders provide stable concentrations after less than two weeks. The
$\delta^{13}C(CO_2)$ isotope ratios were analyzed at the University of Bern (Switzerland) against JRAS-06 reference gases (Wendeberg
et al., 2013; Van Der Laan-Luijkx et al., 2013).

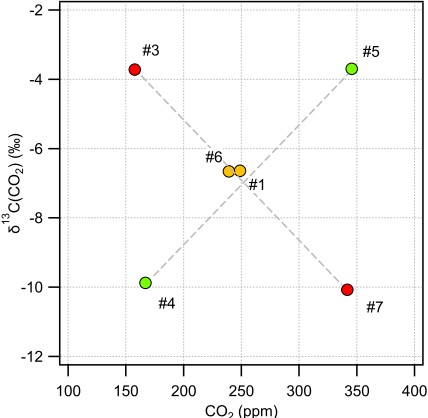

**Figure 6.** Range of $CO_2$ concentration and $\delta^{13}C(CO_2)$ spanned by our custom-made standard gases. The $CH_4$ and $N_2O$ content is indicated
by the different colors: red, green and yellow representing high-, medium-, and low-concentrations, respectively (see also Table 2).

## 3 Performance and Calibration

### 3.1 Precision and stability in flow-through mode

The precision and long-term stability of the system is derived with a reference gas continuously flowing through the instrument. A standard gas (#6 in Table 2) was continuously supplied to the MPC at a constant pressure of $5\,\mathrm{mbar}$ (corresponding to a cell volume of about $1\,\mathrm{ml\ STP}$) and a flow of $7\,\mathrm{mL\,min^{-1}}$ over twelve hours. In this regime, potential artefacts due to sorption effects on surfaces are mostly negligible. Figure 7 shows the results of an Allan-Werle variance analysis (Werle, 2011) of the time-series for all four parameters. The data follow a white noise behavior for about $100\,\mathrm{s}$, except for $CO_2$, which flattens out already after $10\,\mathrm{s}$, and then stays at low values until at least $400\,\mathrm{s}$ before drifts begin dominating the measurements. This information is crucial for an accurate calibration, because it determines the longest time available for a measurement cycle, i.e. the time interval within which a sequence of a discrete air-sample and a subsequent, pressure-adjusted standard-sample has to be analyzed. Currently, our custom-made MPC requires about $300\,\mathrm{s}$ to be completely evacuated and subsequently filled with another gas (see Sect. 3.2). This relatively rapid exchange time is due to the improved leak-tightness, surface inertness as well as lower dead-volumes, and it is well within the constraints defined by the Allan variance minimum. Follow-up experiments using static (batch-mode) configuration led to comparable precision levels to that of flow-through measurements. However, when performing Allan-Werle deviation analyses for discrete samples, the maximum duration of the analyses was limited to half an hour to avoid any noticeable contribution from surface effects related artifacts.

In parallel to the measurements in the MPC, data from the reference cell were also recorded. For $CO_2$, $CH_4$ and $N_2O$, only a marginal correlation between the two time-series was found. Thus, normalization with the reference cell did not improve the performance. The situation is slightly different for $\delta^{13}C(CO_2)$, where the Allan-Werle deviation minimum could be improved towards longer integration times ($>100\,\mathrm{s}$) by using the reference cell records to apply a drift correction in the form of a simple subtraction of the reference cell mean-normalized data from the MPC data. This different behaviour of $\delta^{13}C(CO_2)$ can be understood by considering the facts that i) this quantity is a ratio between two measured quantities of $^{13}CO_2$ and $^{12}CO_2$, and therefore, the random errors of these quantities sum up in the ratio, but correlated drifts are eliminated, and ii) $\delta^{13}C(CO_2)$ is temperature sensitive as discussed in Sect. 2 in the context of line selection. The reference cell data can, therefore, be used for monitoring purposes, i.e. to efficiently flag out periods with daily variations that eventually can happen and (optionally) to further improve the precision of $\delta^{13}C(CO_2)$.

Overall, the precision targets for all parameters are reached in the flow-through mode within $10\,\mathrm{s}$ integration. Integrating over $100\,\mathrm{s}$, the instrument achieves a precision of $0.4\,\mathrm{ppb}$, $0.1\,\mathrm{ppb}$, $0.006\,\mathrm{ppm}$, and $0.02\,‰$ for $CH_4$, $N_2O$, $CO_2$, and $\delta^{13}C(CO_2)$, respectively.

### 3.2 Repeatability of discrete 1 mL STP samples

The final application of the instrument is the accurate (SI-traceable) measurement of discrete, 1 mL air samples that need to be measured in batch mode due to their low volume. As mentioned above, the Allan-Werle deviation analysis indicates that for highest accuracy the sample-standard measurement should take place within $400\,\mathrm{s}$. The gas inlet system (see Fig. 5) was

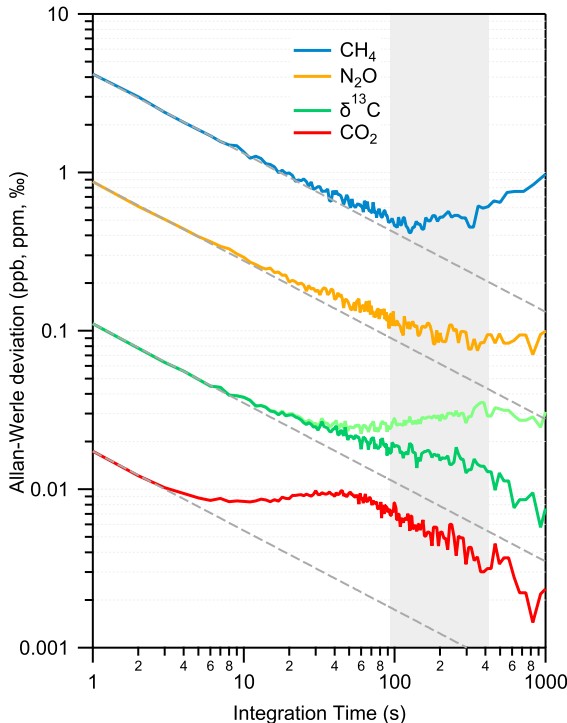

**Figure 7.** Allan-Werle deviation plot for all observed parameters in a flow-through experiment using the standard gas #6. For $\delta^{13}C(CO_2)$, two situations are shown: i) no drift correction (light green) and ii) applying a drift correction based on the reference cell (dark green). While in the case of isotope ratio values a further improvement towards longer integration time can be observed, the concentrations are only marginally influenced (not shown). The optimal integration time is indicated by the gray area.

developed to allow the corresponding rapid switching between gases, while maintaining a precise pressure matching within

$\pm 0.04\,\mathrm{mbar}$. For the removal of the gas from the MPC we also consider the specific ice core application, which foresees the option to cryogenically recollect the gas sample after the spectroscopic measurement in a cold finger for further analysis. Therefore, the sample gas is not simply flushed out of the cell with the stream of the following standard gas, but rather we evacuate the cell using the turbo-molecular pump to levels near to the detection limit of the cell pressure gauge ($10^{-3}\,\mathrm{mbar}$), which is similar to the case of a cryogenic recollection. The total delay time between the sample and the standard is around

$270\,\mathrm{s}$, which is acceptable regarding the stability of the system (Fig. 7) and our precision targets (Sect. 2).

To evaluate the repeatability of discrete $1\,\mathrm{ml}$ sample measurements, the following procedure is used: First, a dip tube is filled with about $1\,\mathrm{ml}$ STP of a standard gas, and then expanded into the evacuated MPC and treated as the 'sample' in the evaluation procedure. Second, for the 'standard' measurement, the MPC is evacuated again, and the same standard gas is introduced directly, i.e. not via dip tube, until reaching the same cell pressure. This sequence takes $10\,\mathrm{min}$ and is repeated several times

$(50\times)$ and consecutively applied for low-, mid- and high-concentration trace gas standards (as indicated in Fig. 6). To minimize biases originating from long-term optical drifts, we recorded spectra with evacuated cell ('zero trace') at the beginning of each

sample-standard pair, which were then subtracted from consecutive spectra. This strategy efficiently removes residual structures in the spectrum that can slowly vary with time and influence the spectral fit.

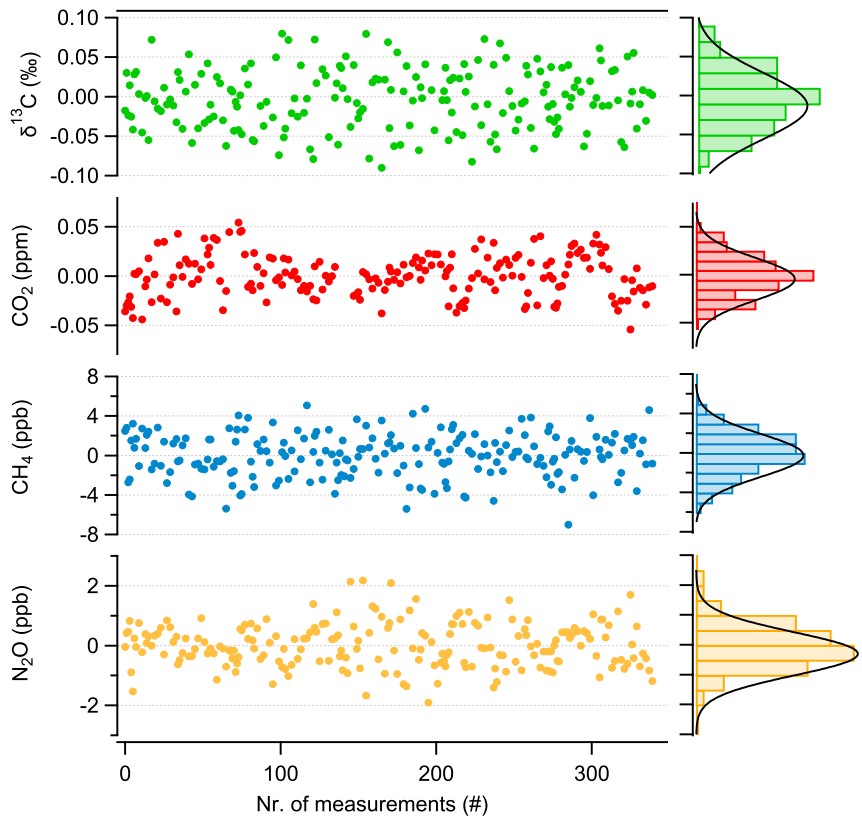

**Figure 8.** Repeatability of discrete $1\,\text{mL}$ sample measurements (mean value subtracted) with associated histograms indicating a normal distribution of the values.

The result of this experiment is summarized in Fig. 8. Here, the averages of individual sample-standard pairs recorded over
$24\,\text{h}$ are displayed along with their associated histograms showing the distribution of the respective values. We found a $1\,\sigma$ standard deviation of $2.2\,\text{ppb}$, $1\,\text{ppb}$, $0.03\,\text{ppm}$ and $0.04\,\text{‰}$ for $CH_4$, $N_2O$, $CO_2$ and $\delta^{13}C(CO_2)$, respectively. These values are higher by a factor 2 for $\delta^{13}C(CO_2)$, 5 for $CH_4$ and $CO_2$, and 10 for $N_2O$ than those derived from the Allan-Werle deviation analysis of the flow-through measurements (see Fig. 7). This indicates that significant uncertainty is introduced by the gas handling process (e.g. pressure and temperature induced instabilities during evacuation/filling process, and/or adsorp-
tion/desorption effects along the sampling system). Although, individual or discrete gas samples can be measured with a precision as determined by the Allan-Werle deviation analysis, additional artifacts/biases, mainly originating from gas handling, will affect the repeatability and accuracy of the measurements when switching between sample gases collected in dip-tubes. Thus, the overall accuracy is not limited by the purely spectroscopic performance, indicating that further improvements may

be possible by optimizing the sample handling. Nevertheless, Fig. 8 clearly demonstrates that with our cyclic measurement
approach, i.e. quickly alternating between dip-tube sample and reference gas measurement, the instrumental drifts can be accounted for and thus, their effect can be minimized even for longer periods of time, i.e. over 24 h. Thus, the target performances are achieved for all compounds, even for discrete samples of only 1 ml STP volume.

## 3.3 Calibration characteristics of discrete 1 mL STP samples

In this section, we investigate the characteristics of the instrument response using the custom-made and externally calibrated standard gases (Table 2) and derive corresponding calibration functions for discrete 1 ml STP air samples. We use the same measurement procedure as described in Sect. 3.2. The only difference is that the sample gas is not transferred via the dip tube, but introduced directly into the MPC in the same way as the standard gas. In other words, the sample-standard pair are basically identical measurements with the same gas at the same pressure (5 mbar). Thus, half of the data were used to monitor potential drifts in the spectrometer response, while the other half served to retrieve the target parameters. First, the values of each category ('anchor' and 'sample') are averaged for 100 s and then, whenever long-term drifts are larger than $3\sigma$ of the uncertainty of individual values, the drifts are taken into account by applying a smoothing-spline (Igor Pro v8, Wavemetrics Inc., USA) over all 'anchor' values considering their individual standard deviation as weighing factor for the spline, and setting the smoothing factor to unity. After drift correction, the ten individual 'sample' values of each standard are averaged and plotted against their reference value, as shown in Fig. 9. Thereby, we found that a linear calibration function describes properly the instrument response for all three trace gas concentrations. Note that the values reported for the laser spectrometer are purely based on spectroscopic (HITRAN) and physical parameters ($P$, $T$, and $OPL$) without any additional calibration. In the case of the $\delta^{13}C(CO_2)$, the following additional approach was used: the $CO_2$ isotope ratio was re-scaled to the VPDB-scale using an isotope ratio value of $R_s = 0.0111802$ (Werner and Brand, 2001). To make the range of measured- and the reference $\delta$-values comparable, an offset of 9‰ was added to the measured values. This offset most likely originates from the uncertainty of line strength of the $CO_2$ isotopologues, but fitting errors (e.g. absorption profile mismatch) or other spectral artefacts can also not be excluded. This offset can optionally be included in the calibration function, i.e. in $d_0$ in Eq. 2.

The calibration of $\delta^{13}C(CO_2)$ is a two-step process as its concentration dependence must also be considered (see e.g. Tuzson et al., 2008). This concentration dependence of $\delta^{13}C(CO_2)$ is determined based on the measurements of standard gases #3 and #5 (see Table 2). As both standard gases have the same $CO_2$ isotopic composition, the apparent difference observed in the spectroscopically-derived isotope ratios is attributed to the difference in the $CO_2$ concentration. This two-point calibration, which was found to be about $0.005$‰ $ppm^{-1}$, is then applied to all the other $\delta^{13}C(CO_2)$ values obtained for the different standard gases. Thus, the specific calibration functions of all four parameters are defined as follows:

$$[X]_{cal} = a_0 + a_1 \cdot [X]_{meas} \tag{1}$$

$$\delta^{13}C_{cal} = d_0 + d_1 \cdot \delta^{13}C_{meas} + d_2 \cdot [CO_2]_{meas} \tag{2}$$

where, the subscripts '*meas*' and '*cal*' denote the measured and the calibrated values, [X] stands for $CO_2$, $CH_4$ and $N_2O$ concentrations, while $a_i$, and $d_i$ represent the fit coefficients. Figure 9 shows the calibration functions and the resulting residuals for all four parameters.

Although, the spectroscopically-retrieved concentrations show a good linear correlation with the reference values, a systematic underestimation is observed, which in case of $CO_2$ is 2.7 %. However, this is well within the total uncertainty of spectral 375 (e.g. line strength) and physical (pressure, temperature, and OPL) parameters. Additionally, there are two other effects that could contribute to the observed bias: i) the Voigt profile used to fit the absorption lines tends to underestimate the effective absorption line areas especially at low pressure, and ii) the absorption lines might be slightly saturated by the incident laser optical power. Underestimation by the Voigt profile is well documented and the literature suggests a bias between 1–2 % for our working pressure (Bui et al., 2014; Lisak et al., 2015). Using more sophisticated line profiles, such as the Hartmann-Tran 380 profile (HTP) (Ngo et al.) in the spectral evaluation has the potential to reduce the observed mismatch (see the characteristic residual structure in Fig. 2), but most likely will not affect the precision or its impact will only be marginal. One key issue is, however, that the additional line parameters required for the HTP line shape model are not available. Nevertheless, the raw spectra have been recorded and they can be post-processed at a later time. Furthermore, there is a strong evidence that optical saturation has a bias contribution as well, despite the implemented intensity reduction scheme (as discussed in Sect. 2.1). 385 This is supported by the observation that increasing cell pressure leads to a decreasing bias in the retrieved concentrations (see Fig. 10), which is in line with the expected behavior of optical saturation; higher gas pressure means higher collision rate, which causes faster relaxation of the states and ultimately reduces the number of molecules that are in the excited state. Based on these experimental data and further investigations of absorption signals at high- and low laser intensities, we estimate, assuming the inhomogeneous broadening regime, the saturation coefficient $s$ to about 0.014, which correspond to a 390 saturation effect of 0.7 %. In any case, the systematic error caused by this effect is included in the calibration process, because the main factors that have an impact on saturation are kept constant, e.g. temperature, pressure, laser intensity or gas matrix. To verify this assumption, we determined the instrument response to our standards in five sets of experiments, with at least two-weeks or more time lag between each. Each set of experiments involved repeated evacuation/filling cycles of the MPC, and every standard gas was measured twenty times in a scheme as described above. We found excellent agreement across all 395 these measurements, which demonstrates the constant analytical performance of the instrument.

It should be noted that the amount of gas resulting from ice core samples, and thus the sample pressure in the MPC, may slightly fluctuate, depending on the air content of the sublimated ice core. At the high level of required accuracy, this may be a critical parameter. Preliminary tests with the sampling and calibration procedure described above were performed to investigate the influence of gas pressure variation on the spectroscopically-retrieved values. For this, we repeated the same procedure that 400 was used for the calibration measurements with the only difference that this time the 'sample' measurements were done at various pressures covering the range between 2 and 10.5 mbar. Figure 10 shows the representative results for one standard gas. This indicates that basically every target parameter is affected by changing cell pressure, despite the fact that the gas pressure values are used in the fitting algorithm to calculate the corresponding Lorentzian width contributions to the absorption line and also in the determination of the number density of the target species based on the ideal gas law. Whilst for the $CH_4$ and $N_2O$

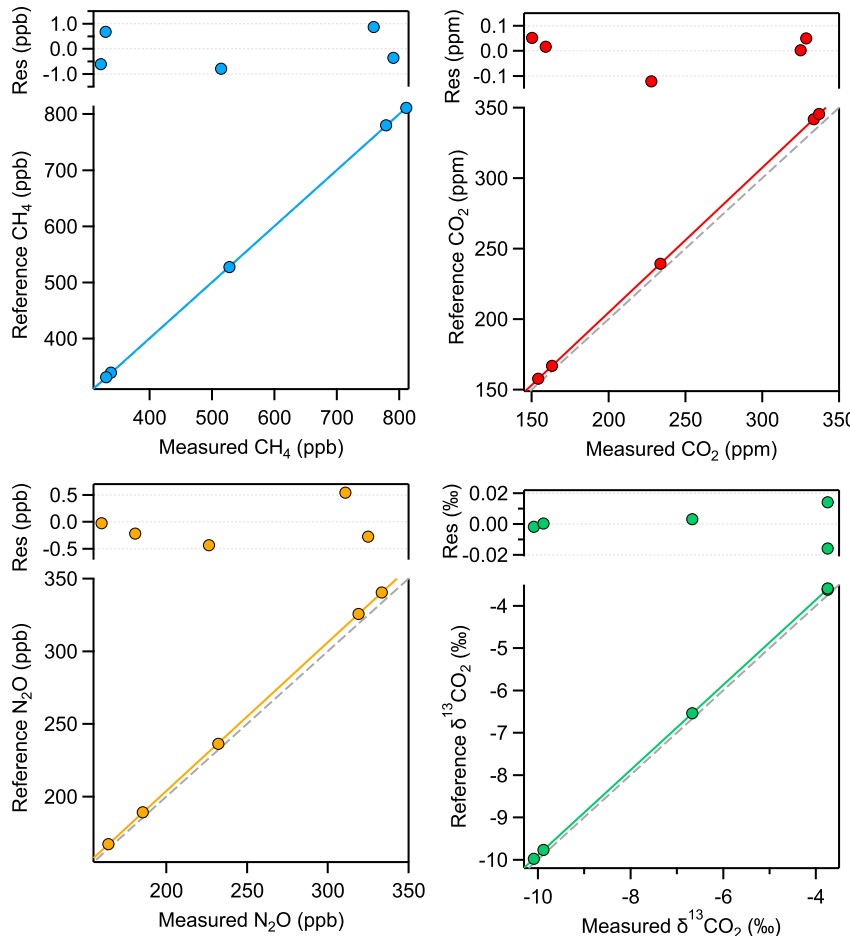

**Figure 9.** Calibration functions and corresponding residuals from the fit of all four analyzed parameters for a sample of $1\,\mathrm{ml}$ STP volume, corresponding to $5\,\mathrm{mbar}$ pressure in the MPC. The dashed gray lines indicate the 1:1 correlation as reference.

concentration retrieval the influence of pressure is about $0.76 \pm 0.19\,\mathrm{ppb\,mbar^{-1}}$ and $0.31 \pm 0.03\,\mathrm{ppb\,mbar^{-1}}$, respectively, the $\delta^{13}\mathrm{C}(\mathrm{CO_2})$ and the $\mathrm{CO_2}$ concentration values show a strong and non-linear dependence, especially towards lower cell pressures ($<5\,\mathrm{mbar}$). Obviously, the pressure dependence can deteriorate the spectrometer accuracy if it is not taken into account properly. Considering a narrow pressure region around $5\,\mathrm{mbar}$, the dependency of $\delta^{13}\mathrm{C}(\mathrm{CO_2})$ on pressure is about $0.81\,‰\,\mathrm{mbar^{-1}}$. Thus, in order to achieve $0.04\,‰$ precision on the isotope ratio, an accuracy of $0.05\,\mathrm{mbar}$ on the pressure

measurement is needed. Our pressure sensor (CMR 363, Pfeiffer Vacuum, Switzerland) has a stated accuracy of $0.2\,\%$ of the measured value, i.e., an uncertainty of $0.01\,\mathrm{mbar}$ for the $5\,\mathrm{mbar}$ sample pressure. As mentioned above (see sect.2.3), the pressure matching between consecutive batch samples is better than $0.04\,\mathrm{mbar}$. Based on these findings, the pressure either has to be actively controlled and accurately adjusted or it has to be considered in the data retrieval. In the former case, an additional

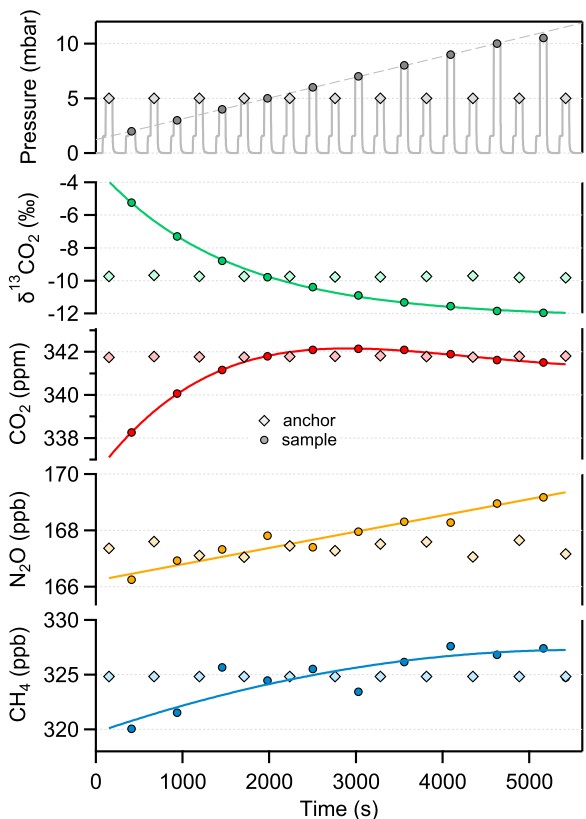

**Figure 10.** Pressure dependence of the target parameters. The same standard gas is used alternately as 'anchor' (diamond) and 'sample' (circle). The gas pressure for the 'anchor' is kept at $5\,\mathrm{mbar}$, while the pressure of the 'sample' is gradually changed between 2 and $10\,\mathrm{mbar}$ (top plot). The markers represent the mean value of the measurements taken at each pressure value over $70\,\mathrm{s}$. The overall sequence is repeated three times for the same standard gas and then consecutively applied to the other five standards. The instrumental response remained stable during the entire measurement time ( $>25\,\mathrm{h}$).

buffer volume with flexible bellows can be added to the gas-handling system to precisely match the cell pressure to a preset
value for any ice core sample. Otherwise, the calibration functions have to be known for each pressure value. This is feasible because the pressure dependency of the target parameters have smooth characteristics and can be described by simple analytical functions (linear, polynomial or exponential). These relations hold also for the calibration function coefficients, which can then include a pressure-dependence correction term. Although, it involves a slightly higher complexity, such an approach was tested successfully using interpolated calibration function coefficients for a randomly taken pressure value. Nevertheless, the final
accuracy of the measurement will reflect the uncertainty of the extrapolation between the used individual calibration functions. For the ice core application, a more detailed characterization and validation of the pressure-dependent calibration parameters in a smaller pressure range around the pressure span expected from an ice core sample is required. Here, we provided the instrument behavior over the whole possible operation range, which is the basis for future work in its final configuration

including the ice core extraction system. Furthermore, it is advisable to use standard gases that are similar to the unknown
sample in both concentration and isotopic composition. Last but not least, applying a more sophisticated line shape model in
the fitting algorithm (see discussion in Sect. 3) has also the potential to minimize the observed pressure dependence by better
matching the measured absorption signal.

Finally, a significant advantage of laser spectroscopy over - for instance - mass spectrometry is that the absorption lines are
molecule-specific and interferences between different gases are unlikely. In the case of ice core analyses, in particular, the use
of organic drill fluids may, however, lead to contamination with potentially absorbing compounds. Therefore, we tested for
spectral interferences with the drill fluid ESTISOL™ 140, which is to be used in Antarctica within the Beyond EPICA ice
core project. This substance was introduced into the multipass cell in trace quantities that are expected to be representative
for the ice core samples. This led to no alterations in the observed spectrum within the selected spectral window, which is
not surprising, because such large organic compounds usually have very broad absorption features and tend to be localized to
frequencies typical for functional groups.

## 4   Conclusions

A dual-QCL direct absorption spectrometer has been developed for the challenging measurements of small ($1\,\mathrm{ml}$ STP) ice core
samples. The stringent requirements in precision and accuracy were met by the design, development, and implementation of
the following key elements: i) a new low-volume multipass absorption cell was designed with a special focus on minimizing
contamination effects (e.g. surface adsorption, outgassing, leak-rate), improving optical performance (low fringe level) and
robustness against mechanical and thermal variations; ii) a custom-made low-noise/low-drift laser driving electronics stabilized
by a thermoelectrically cooled water-loop; iii) a fast dual-channel real-time data acquisition system based on FPGA SoC; and
iv) a dedicated system-tailored and automated gas handling manifold to manipulate low-volume samples.

Flow-through experiments demonstrate an analytical precision ($1\,\sigma$) of $0.006\,\mathrm{ppm}$ for $CO_2$, $0.02\,‰$ for $\delta^{13}C(CO_2)$, $0.4\,\mathrm{ppb}$
for $CH_4$ and $0.1\,\mathrm{ppb}$ for $N_2O$ obtained after an integration time of $100\,\mathrm{s}$. Sample-standard repeatabilities ($1\,\sigma$) of discrete
samples of $1\,\mathrm{ml}$ STP amount to of $0.03\,\mathrm{ppm}$ for $CO_2$, $0.04\,‰$ for $\delta^{13}C(CO_2)$, $2.2\,\mathrm{ppb}$ for $CH_4$ and $1\,\mathrm{ppb}$ for $N_2O$ and
meet or even exceed our ice core precision targets. Furthermore, calibration curves have been determined and verified in
repeated measurements over a time span of several months, and it was found that the instrument provides calibrated values
with uncertainties similar to the repeatabilities. Thus, the spectrometer is capable of simultaneously and accurately analyzing
discrete air samples of $1\,\mathrm{mL}$ STP volume for their $CO_2$, $\delta^{13}C(CO_2)$, $CH_4$ and $N_2O$ composition. Further improvements of
the instrument performance can be expected from the implementation of a more sophisticated line profile beyond the Voigt-
model in the spectral fitting algorithm. Thereby, a more accurate concentration retrieval and potentially a less eminent pressure
dependence may be achieved, which would alleviate the pressure on careful calibration as used in this study. Finally, it was
verified that the laser spectroscopic approach is immune against potential contamination compounds such as drilling fluids.
Being a non-invasive technique, it allows reusing the precious samples for further ice core analyses after cryogenic recollection

from the multipass cell. Overall, this approach opens many options for further analytical improvements and technological developments in ice core research.

*Data availability.*  The data used in this manuscript are available from the corresponding author upon request.

*Author contributions.*  B.B. designed and developed the instrument under the guidance of B.T. and with input by L.E. and H.F., while L.M.,
D.B., and J.S. realized the gas handling system. B.B., L.M., D.B., and J.S. performed the experiments and evaluated the data. P.S. designed and developed the electronics hardware. A.K. developed and implemented FPGA and DAQ functionalities. H.L. developed the spectral analysis and hardware control software. H.F together with L.E. and B.T. designed the research, managed and supervised the project, and discussed the results. B.B. and B.T. prepared the manuscript with contributions from all authors.

*Competing interests.*  The authors declare that they have no conflict of interest.

*Acknowledgements.*  We acknowledge Nicolas Sobanski and Oleg Aseev for their support in designing the multipass cell mirrors. We thank Erich Heiniger from the Empa workshop for the design and construction of the MPC. Christoph Zellweger and Martin Vollmer are acknowledged for supporting and supervising the preparation of the custom-made standard gases. Markus Leuenberger and his team from University of Bern (Switzerland) is acknowledged for performing the IRMS measurements on our custom-made standards. This study is part of the ERC Advanced Grant "deepSLice" (667507) of H.F. This project has received funding from the European Research Council (ERC) under
the European Union's Horizon 2020 research and innovation programme (grant agreement No 667507).

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
