# Peer review of "High-precision laser spectrometer for multiple greenhouse gas analysis in 1 mL air from ice core samples"

_Atmospheric Measurement Techniques, 2020_

## Referee Comment (RC1) · Anonymous Referee #1 · 11 Sep 2020

The broad research question behind this paper – how can the tiny volumes of gas contained within the heavily thinned 'Oldest Ice' be analyzed effectively without sacrificing data quality – is of interest to many in the paleoclimate community. Unfortunately, despite the promising title, abstract and introduction, this study provides more questions than answers. The reader is not told until L53, after an Introduction loaded with information related to the Oldest Ice challenges, that this study only reports on the development of a new laser spectrometer for multiple gas analytes. This study does not include any information about the sublimation technique, i.e., how the gas sample will be extracted from the ice. This makes it impossible to assess the potential efficacy of the system for ice core measurements, which makes the last sentence (L377) seem a

little premature, and the title, abstract and introduction seem misleading. I don't mean to criticize the significant achievement of the authors in developing this instrument, just to highlight that the instrument is a long-way from being well-suited to ice core analysis.

In particular, the section beginning at L337 focuses on one of the challenges of using laser spectrometers for small gas volumes – that the volume of gas must be kept constant. Figure 10 nicely shows the problems this causes for the measurements. As the sublimation system is not described and no results reported, it is difficult to judge how effectively this problem will be dealt with via the calibration method suggested.

I am not an expert in optical physics but, from what I understand, the instrument design is well-described and includes enough novel features or improvements on previous work to warrant a publication. At two points I felt that more technical information could have been provided: L83 on the selection of absorption lines, and ~L102 on the beam characterization.

The figures are all clear and helpful and the paper itself is well-written.

Other comments: Abstract L10: Where are these precision values from? I can't find their origin in the manuscript. Values at L288 are different. Please state that these are 1 sigma values, if that is the case.

L11: How do "Repeated measurement cycles of air samples" demonstrate "an excellent accuracy level"?

Introduction: Nice summary of modern measurements and the Oldest Ice campaign but it doesn't provide much context for this technical study on spectrometer design. L68-71: Both studies cited here use Picarro instruments that do not utilize cavity-enhanced technology. L77-78: How were these values chosen? Are they 1 sigma values? L151: Last sentence is repetition.

Section 3.1, L255, sentence beginning "Our setup..." I didn't catch the meaning here. L258: Can the authors further explain why d13C-CO2 and CO2 behave differently to

the other analytes during the Allan-Werle test? Why do the d13-CO2 measurements drift significantly when the others do not?

L270-271: The Allan-Werle shows the optimum time to integrate the data over to obtain high precision. How does this ensure high accuracy?

Figure 9 and associated text: Much of this underestimation of all the analytes is attributed to optical saturation. . .this seems to be a significant problem because the offset changes with pressure. What is preventing you from reducing the intensity further? The 'neutral density filter' mentioned at L118 doesn't seem enough here.
* * *

---

## Referee Comment (RC2) · Anonymous Referee #2 · 14 Sep 2020

Review of AMT-2020-279 manuscript

General comments:

The work by Bereiter et al. "High precision laser spectrometer for multiple greenhouse gas analysis in 1 mL air from ice core samples", describe the development of a new optical spectrometer for multiple detection of $CH_4$, $N_2O$, $CO_2$ and $d13CO_2$ using a dual-QCL multipass absorption technique. The manuscript is clear and well written, the authors did a good effort on the development as well as on the instrument characterization. This sensor is unique on its kind since it is optimized for small samples size (with the aim of using it for ice core measurements) and include the high precision

measurement of d13CO2 which is key on the study the climate / carbon cycle coupling. I believe that the performance of the spectrometer could have been improved by improving the fit routine, therefore I suggest to the authors to explore this suggestion. This hopefully would not require them to redo the analysis, if they saved data for playing with the fit parameters at posteriori. However, I would say that this is not compulsory since the authors seems to have achieved the target precision on the measurements (although it would further improve the precision, and it would perhaps allow to remove or at least minimize the corrections on the pressure and concentration dependencies). I therefore recommend this manuscript for publication on AMT, after considering the comments below.

Specific comments:

The whole story regarding the absorption line saturation need more discussion in the paper. Authors should mention the optical power in the multipass cell as well as the beam waist. With those two parameters, the optical pathlength and two parameters available from the HITRAN database (the Einstein coefficient and the degeneracy of the excited state) they should be able to estimate the saturation parameter (s), which is the ratio between the intensity (I) used and the saturation intensity (I_sat) that is calculated. Then, the difficulty to find the exact saturation effect is due to the fact that it depends on the regime "homogeneous" (Lorentzian broadening dominate) or "inhomogeneous" (Gaussian broadening dominate). But authors could calculate this assuming that at 5 mbar they are in the inhomogeneous regime. Authors should therefore calculate their saturation intensity (I_sat in W/cm2) and s = I/I_sat, and therefore the effect of saturation on the absorption lines (1/sqrt(1+s)). An s of 0.05, for instance would corresponds to a 2.5% effet on the absorption lines. Then they should check if this value can be in line with an estimation of the amplitude stability of their laser source. (A good reference for saturation spectroscopy is Giusfredi, et al.: Theory of saturated-absorption cavity ring-down: radiocarbon dioxide detection, a case study, J. Opt. Soc. Am. B, 32(10), 2223, doi:10.1364/josab.32.002223, 2015).

L51-52. In the paper you mention about discrete measurements in ice-core and here you mention a continuous extraction system. Please be more consistent on this point. And if possible, please refer to a paper "in preparation" for the extraction technique.

L84-85. In the selection of the absorption lines did authors toke into account the temperature dependency of the absorption lines? This should be then discussed in the manuscript.

Figure2. This wiggled shape in the residual is very large, I think authors should consider using a different line-shape profile which would include Dicke narrowing and perhaps also speed dependence collisional broadening. The intensity of the peak absorption is only 150 time higher than the peak-to-peak on the residual, which is in with the 11 permil precision on a single spectrum. But this also show that on the d13CO2 precision authors are limited to those wiggles and that they could further improve their measurements. Furthermore, looking at HITRAN2016 simulation the intensities at 5mbar and 20°C do not seems to have the same intensity as showed here. Is there an error in the HITRAN database? If so, authors could mention it in the paper and say which line has the wrong cross-section and by how much is wrong (based on their gas standard mixtures). Last, they should be mention that this is a single scan spectrum (acq. Time 75 ïA■s). What is the strategy then? How many spectra the authors average before perform the fit? This should be also mentioned.

L116-119. Please provide the power injected in the MPC. Are authors not annoyed by optical fringes from the ND filter?

L121: how the accuracy on the temperature was measured? With an independent temperature probes? is this the accuracy of the temperature stabilization or just the precision of the temperature sensor itself? Please specify. This could then be related to the temperature dependency of the absorption lines and to the achieved precision on the d13CO2.

L130-131. What about for the d13CO2? It is weird that author fixed they required

optical pathlength based on the CH4 and N2O required precision rather than on the d13CO2 measurement.

L139. Significantly longer. Could you provide a value?!

L155-156: How authors can simulate the frequency of the optical fringes produced by the multiple passages?! It is just a maximization of the distance between the neighboring spots as mentioned few lines below or is something else?

L203. Critical orifice. Can authors mention the size? Well, in flow measurement this would probably fractionate... please explain.

Table 1. When second digit precision is reported (eg. 0.01) then concentration should be reported with the same number of digits (eg. 248.8x ppm). Did authors monitor their standard mixtures over time? Do they see any drift?

Figure 7. It would be interesting to see the same AW statistical analysis for a static experiment. Can authors add this information? Since the ice core measurements will certainly done using discrete samples and in static mode, I guess.

L281: This sequence is repeated several times: Can authors proved information about the frequency of those 50 measurements?

L293-294. "Even for discrete samples... ". What do they mean with this sentence? Because for a single discrete measurement the performances are maintained for a time given by the AW analysis, which is shown only a maximum integration time of 16 minutes. What authors mean with this 24h of continuous operation? Can they prove it with an AW plot?

L311-312: "This two-point calibration... " This is normally not a linear relationship, but rather an exponential with discrepancy becoming larger and larger at low concentrations... It is possible that authors would be able to approximate it to a linear relation in the range of concentrations they will have in the ice cores? Why authors did not just apply different dilution factors to on of their standard to study this effect on

a larger concentration range? Just to mentioned, I see an artefact of 0.94 permil on the working range of 157.7 - 345.5 ppm, which is more the 20 times larger than the claimed accuracy.

Figure 9. So, if I understand well, the difference between the measured and the reference corresponds to the difference between the measurement done by the commercial instruments used for calibrate the standard gas mixtures and your new instrument, is that correct? But you should say what is your approach for your instrument. Do you stick with the intensities provided by HITRAN database or you calibrate the spectrometer with one of the standard bottles? Because if it is the second option, then you should have at least one point which would match well the reference measurement. And what about the d13CO2? Because for that you have to rely on a reference measurement. So I do not get why there is this offset between reference and measurement for all isotopic ratios rather than a crossing point.

L343. "This indicates that. . ." You should explain your approach here. I believe that you leave the Lorentzian contribution on the absorption lines free to adjust with pressure. This should be mentioned in the text. As well as that the concentration in mixing ratio is then corrected for the total gas pressure in the cell. I believe that the artefact you see could be removed (or at least minimized) by changing the fit profile as mentioned above.

L345. Can authors estimate the dependency for d13CO2 around 5 mbar? what precision in pressure is required the 0.04 permil precision and how it compares with your accuracy on the pressure measurement?!

L356: "possibly at value >= 5 mbar." Here you should rather report 5 +/- xx mbar, with the error estimated in order to stay within the 0.04 permil (comment above).

L370-371: mention that those are 1 sigma precisions.

Technical corrections:

L9. I would rather call it multi-pass absorption rather than direct absorption (or direct multi-pass absorption if authors want to keep the word direct).

L61-80. This part should go to the introduction rather than in the methods. The latter should be focus on the description of the methods.

L108. Mention the photodiode model.

L173. I suggest By applying.

L293: replace "targets" with "target performances"
* * *

---

## Short Comment (SC1) · 14 Sep 2020

This is a great study and a great instrument. I found the paper very interesting to read throughout but had one question constantly in the back of my mind: how will the instrument deal with the presence of ice core drilling fluid in the gas sample? Of course, we do everything we can to avoid this and there are ways to mitigate the problem (that are beyond the scope of this study). I was thinking more along the lines of whether this was considered during the design phase. Or if the author's have been able to do theoretical or experimental work on potential spectral interferences. That's all.

If drilling fluid can be shown not to be a problem, than the method will have (another)

major advantage on dual-inlet IRMS measurements of carbon isotopes as drilling fluid can be the Achilles' Heel of this technique. However, if drilling fluid is problematic, it's not clear to me if traditional GC-MS techniques remain better suited for small samples.

All the best, Thomas Bauska

---

## Author Comment (AC1) · 22 Oct 2020

The comment was uploaded in the form of a supplement:
https://amt.copernicus.org/preprints/amt-2020-279/amt-2020-279-AC1-supplement.pdf

---

## Author Comment (AC2) · 22 Oct 2020

The comment was uploaded in the form of a supplement:
https://amt.copernicus.org/preprints/amt-2020-279/amt-2020-279-AC2-supplement.pdf

---

## Author Comment (AC3) · 22 Oct 2020

Thank you for your thoughts and the relevant question. Yes, we are well aware of this highly challenging issue and therefore we clarified beforehand the ex-pected type of the drilling fluid that will be used within the "Beyond EPICA Oldest Ice Core" project. We will add the following lines to the manuscript: "A significant advantage of laser spectrometry over - for instance - mass spectrometry is that the absorption lines are gas-specific and interferences between different gases unlikely. In the case of ice core analyses, in particular, the use of organic drill fluids may lead to contamination with potentially absorbing trace gases. Therefore, we tested for interferences with the drill

fluid (ESTISOL^TM 140) which is to be used in Antarctica within the Beyond EPICA ice core project. We purchased pure ESTISOL^TM 140 and introduced the headspace of drill fluid contaminated sample into the multipass cell in quantities that we considered representative for the ice core samples. This lead to no alterations in the spectrum within the wavenumber window covered by our two lasers. The main reason is that these large organic compounds have very broad absorption features and tend to be localized to frequencies typical for functional groups. Therefore, our narrow-band laser spec-troscopic approach in the mid-IR is not affected by such contaminations in the studied wavenumber range."

---

## Author Response (AR1)

We would like to thank the Referee for the constructive comments and helpful suggestions on the manuscript, which helped us to further improve the clarity of the paper. Below, we give detailed responses (in blue) where appropriate.

**Anonymous Referee #1**

**General comments:**

Unfortunately, despite the promising title, abstract and introduction, this study provides more questions than answers. The reader is not told until L53, after an Introduction loaded with information related to the Oldest Ice challenges that this study only reports on the development of a new laser spectrometer for multiple gas analytes. This study does not include any information about the sublimation technique, i.e., how the gas sample will be extracted from the ice. This makes it impossible to assess the potential efficacy of the system for ice core measurements, which makes the last sentence (L377) seem a little premature, and the title, abstract and introduction seem misleading. (...) the instrument is a long-way from being well-suited to ice core analysis. In particular, the section beginning at L337 focuses on one of the challenges of using laser spectrometers for small gas volumes – that the volume of gas must be kept constant. Figure 10 nicely shows the problems this causes for the measurements. As the sublimation system is not described and no results reported, it is difficult to judge how effectively this problem will be dealt with via the calibration method suggested.

We regret that the Reviewer was misled by the title/abstract of our manuscript and we apologize for not meeting his/her expectations. In the revised version, we make clear very early on, i.e. in the abstract and the introduction, that this is only the first of two papers on this topic. The second paper will describe the sublimation extraction in detail and will be submitted next year. However, a thorough description of the QCLAS itself, as done in this manuscript, is already a formidable challenge and easily fills one manuscript. The multi-species capability of our unique dual-laser instrument and its unmatched performance for small air samples of only 1 ml STP in batch mode with similar or even better performance than off-the-shelf spectrometers in through-flow mode and high sample pressures, in our opinion represents a substantial scientific progress and easily justifies publication on its own. Regarding the issue of pressure adjustment, we accentuate in the revised manuscript that (i) using our calibration we can correct for the pressure dependence and (ii) that we designed our QCLAS inlet system in such a way that we can adjust the pressure of our standard gases to the pressure of the ice core derived sample within 0.04 mbar. Moreover, our multi-gas calibration routine ensures that we always have a standard gas that is similar in concentration and carbon isotopic signature to the sample derived from the ice cores. This implies that the important issue concerning the size of the ice core sample is already solved.

At two points I felt that more technical information could have been provided: L83 on the selection of absorption lines, and~L102 on the beam characterization. The figures are all clear and helpful and the paper itself is well-written.

Added a table with the information about the selected absorption lines. Values for the beam waist and the optical power were also added to the main text.

**Other comments:**

Abstract L10: Where are these precision values from? I can't find their origin in the manuscript. Values at L288 are different. Please state that these are 1 sigma values, if that is the case.

There was a mistake regarding the precision values for  $N_2O$  and  $CH_4$ . They should read as given at L288 as pointed out by the Referee. Replaced the wrong values and added the information about the  $1\sigma$ .

L11: How do "Repeated measurement cycles of air samples" demonstrate "an excellent accuracy level"?

We realize that this statement was difficult to understand in the given context. We were referring to repeated calibration runs, which agreed very well within the instrument measurement uncertainty. The wording was changed accordingly.

Introduction: Nice summary of modern measurements and the Oldest Ice campaign but it doesn't provide much context for this technical study on spectrometer design.

The instrumental development with its specific design has been fully driven by the ice core project, and it is important that this becomes clear in the introduction. Nevertheless, we are thankful to the reviewer for his feedback, and we revised our manuscript to inform the reader early (both in the abstract and the introduction) that this paper describes the QCLAS development.

L68-71: Both studies cited here use Picarro instruments that do not utilize cavity-enhanced technology.

The Picarro instruments are based on cavity ring down technique, and the signal is enhanced through the cavity length. Therefore, it is correct to refer to as cavity enhanced technology in this context.

**L77-78: How were these values chosen? Are they 1 sigma values?**

Added to the manuscript: " In order to allow for authoritative interpretation of the observed glacial/interglacial changes in the biogeochemical cycles of these three greenhouse gases, a signal-to-noise ratio of better than 5 for the centennial to multi-millennial variations found in ice cores for CO2, CH4, N2O, and  $\delta^{13}C(CO_2)$  over the last glacial cycles is required. This results in precision targets of these parameters for high-quality ice core analyses of 0.5 ppm, 2 ppb, 2 ppb, and 0.04 ‰, respectively. These targets are either comparable or better than the best ice core analysis systems available to date."

**L151: Last sentence is repetition.**

To avoid confusion we rephrased this sentence: "*Thus, this concept allows for a simpler alignment mechanism and hence it also minimizes the dead volume behind the rear-mirror that would be required otherwise.*"

Section 3.1, L255, sentence beginning "Our setup..." I didn't catch the meaning here.

To clarify the text, the following modification was applied: "*Currently, custom-made MPC requires about 300 s to be completely evacuated and subsequently filled with another gas. This relatively rapid exchange time is due to the improved leak-tightness and minimal surface interactions, and it is well within the constrain defined by the Allan variance minimum.* "

L258: Can the authors further explain why d13C-CO2 and CO2 behave differently to the other analytes during the Allan-Werle test? Why do the d13-CO2 measurements drift significantly when the others do not?

The  $\delta^{13}C(CO_2)$  is a ratio given by the two measured quantities of  ${}^{12}CO_2$  and  ${}^{13}CO_2$  and, therefore, the random errors of these quantities sum up in the ratio. Furthermore, the  $\delta^{13}C(CO_2)$  is also temperature sensitive as now mentioned explicitly in Sect 2 in the context of line selection. See also our reply to the comments of Referee #2. To clarify this detail we added the following text:

"This different behavior of  $\delta^{13}C(CO_2)$  can be understood by considering the facts that i) this quantity is a ratio between two measured quantities of  ${}^{13}CO_2$  and  ${}^{12}CO_2$ , therefore, the random errors of these quantities sum up in the ratio, but correlated drifts are eventually eliminated, and ii) the  $\delta^{13}C(CO_2)$  is highly temperature sensitive as discussed in Sect. 2 in the context of line selection."

L270-271: The Allan-Werle shows the optimum time to integrate the data over to obtain high precision. How does this ensure high accuracy?

If one can measure a reference gas right after the unknown gas sample was analyzed and keep this alternating measurement cycle within the time period defined by the averaging time corresponding to

the Allan variance minimum, then the accuracy can be as high as the precision, assuming that the reference is accurately known. In other words, precision is a necessary but not a sufficient prerequisite for accuracy.

Figure 9 and associated text: Much of this underestimation of all the analytes is attributed to optical saturation...this seems to be a significant problem because the off-set changes with pressure. What is preventing you from reducing the intensity further? The 'neutral density filter' mentioned at L118 doesn't seem enough here.

We already reduced the laser initial intensity by a factor of 10. Further reduction would lead to a decrease of the SNR because of detector noise, which then would negatively affect the precision. For the targeted precision, we have to make this compromise between enough signal and potential optical saturation. In any case, our calibration routine takes care of systematic effects introduced by slight optical saturation. These details are now included in the revised manuscript.

**Anonymous Referee #2**

**General comments:**

(...) I believe that the performance of the spectrometer could have been improved by improving the fit routine, therefore I suggest to the authors to explore this suggestion. This hopefully would not require them to redo the analysis, if they saved data for playing with the fit parameters at posteriori. However, I would say that this is not compulsory since the authors seems to have achieved the target precision on the measurements (although it would further improve the precision, and it would perhaps allow to remove or at least minimize the corrections on the pressure and concentration dependencies). I therefore recommend this manuscript for publication on AMT, after considering the comments below.

We fully share these thoughts with the Referee. Using more sophisticated line profiles in the spectral evaluation has the potential to reduce the current pressure dependence, but most likely the impact will only be marginal. At the moment, however, we don't have a fitting routine going beyond the Voigt-profile, but we are working on implementing the HTP-profile fitting algorithm in the near future. One serious issue here is, however, the four missing line parameters for the molecular transitions. Neverthe-less, the raw spectra have been recorded and they can be post-processed later, once the parametrization is available. This aspect is now included in the conclusions as further potential for future improvement.

**Specific comments:**

The whole story regarding the absorption line saturation need more discussion in the paper. Authors should mention the optical power in the multipass cell as well as the beam waist. With those two parameters, the optical path length and two parameters available from the HITRAN database (the Einstein coefficient and the degeneracy of the excited state) they should be able to estimate the saturation parameter (s), which is the ratio between the intensity (I) used and the saturation intensity (I\_sat) that is calculated. Then, the difficulty to find the exact saturation effect is due to the fact that it depends on the regime "homogeneous" (Lorentzian broadening dominate) or "inhomogeneous" (Gaussian broadening dominate). But authors could calculate this assuming that at 5 mbar they are in the inhomogeneous regime. Authors should therefore calculate their saturation intensity (I\_sat in W/cm2) and  $s = I/I_sat$ , and therefore the effect of saturation on the absorption lines. Then they should check if this value can be in line with an estimation of the amplitude stability of their laser source. (A good reference for saturation spectroscopy is Giusfredi, et al.: Theory of saturated-absorption cavity ring-down: radiocarbon dioxide detection, a case study, J. Opt. Soc.Am. B, 32(10), 2223, doi:10.1364/josab.32.002223, 2015).

We agree that the saturation parameter could be roughly estimated, but this will not help us improving our instrument performance. Based on experimental evidence, we know that for sample pressures above 10 mbar the saturation is negligible (< 0.2 %). The same is true for further reduced laser intensity (<

1mW). However, the gas pressure is defined by the sample availability. Therefore, we expect the gas pressure to be  $\leq$ 5 mbar. Unfortunately, a further reduction of the laser intensity has a negative impact on the SNR, i.e. on the analytical precision (see also our reply to Reviewer #1).

Nevertheless, the well-founded remark of Reviewer 2 motivated us to have a closer look at the saturation effect by (i) comparing absorption profiles, recorded at a fixed gas pressure, but with two different laser intensities, and (ii) by changing sample pressure at fixed laser intensity. From these empirical data, and assuming the inhomogeneous regime as proposed by the Referee, we estimated a saturation intensity and thereof the saturation factor s of 0.014. This effect can account for about 0.7 % reduction in the line absorption. Given the discrepancy between observed and estimated values, we checked the input parameters of the fitting routine and found a mistake in the pressure conversion (hPa to atm). Thereby, a systematic bias of 2.6 % was introduced to the retrieved concentrations. Since this mainly affects the calibration factors, we re-calculated these values and also added a short paragraph about the saturation. The revision in our pressure value and calibration has no further implications for the quality of our measurements.

We are very thankful to the Referee for drawing our attention to this issue.

L51-52. In the paper you mention about discrete measurements in ice-core and here you mention a continuous extraction system. Please be more consistent on this point. And if possible, please refer to a paper "in preparation" for the extraction technique.

Text revised to: "In this publication, we will present in detail the laser absorption spectrometer and its performance, while the continuous sublimation extraction system providing a cm-scale vertical resolution will be described in a separate paper (Mächler65et al., 2020)."

L84-85. In the selection of the absorption lines did authors take into account the temperature dependency of the absorption lines? This should be then discussed in the manuscript.

Yes. Added a short paragraph on this topic in the revised paper.

Figure2. This wiggled shape in the residual is very large, I think authors should consider using a different line-shape profile which would include Dicke narrowing and perhaps also speed dependence collisional broadening. The intensity of the peak absorption is only 150 time higher than the peak-to-peak on the residual, which is in with the 11 permil precision on a single spectrum. But this also show that on the d13CO2 precision authors are limited to those wiggles and that they could further improve their meas-urements. Furthermore, looking at HITRAN2016 simulation the intensities at 5 mbar and 20°C do not seems to have the same intensity as showed here. Is there an error in the HITRAN database? If so, authors could mention it in the paper and say which line has the wrong cross-section and by how much is wrong (based on their gas standard mixtures). Last, they should be mention that this is a single scan spectrum (acq. Time 75 us). What is the strategy then? How many spectra the authors average before perform the fit? This should be also mentioned.

See above our reply to the general comments. Our midterm goal is to implement the HTP-profile fitting algorithm for spectral evaluation. However, this will mainly remove systematic biases, i.e. improving accuracy rather than affecting precision. Furthermore, as we use the integral of the absorption profile instead of the peak amplitude, the situation is much more relaxed as the integral over the residual is approaching zero. The current performance allows us to reach the precision targets, but we have to perform careful calibration to achieve the required accuracy. The information about spectral averaging is given at line L190 and a link to Fig.2 is provided at L196. Added sentence on future potential development in spectral analysis in the Conclusions.

L116-119. Please provide the power injected in the MPC. Are authors not annoyed by optical fringes from the ND filter?

The laser power after the ND filter entering the MPC is 5.4 mW. The ND filter was custom-made using a wedged substrate. This reduces the unwanted etaloning effect and the remaining modulation is as broad

as our spectral coverage. Nevertheless, mechanical and temperature effects still may affect the instrument long-term stability. Therefore, we use a water-cooled optical plate, stabilized at 5 mK, and a hermetically sealed housing around the optical module. We added this information to the manuscript.

L121: how the accuracy on the temperature was measured? With an independent temperature probes? is this the accuracy of the temperature stabilization or just the precision of the temperature sensor itself? Please specify. This could then be related to the temperature dependency of the absorption lines and to the achieved precision on the d13CO2.

The base-plate temperature was measured indirectly based on the T-sensor included in the thermochiller used to circulate the cooling liquid. The gas sample temperature in the MPC was measured by high-precision NTC sensors, which were not calibrated in our laboratory and may have some bias (~0.5 %), while their precision is much better (mK). The temperature effect on  $\delta^{13}C(CO_2)$  can be easily estimated considering the lower state energies of the involved transitions, which results in a temperature sensitivity of ~16 ‰/K. Although it may be difficult to maintain long-term stability of the absorption cell to better than 0.05 K, changes in the cell temperature can be measured with a precision on the order of mK using high-precision thermistors. Since the measured temperature is used continuously for interpreting the absorbance spectra, the temperature dependence of the line strength is not a major impediment to obtain isotope ratio precisions better than 0.02 ‰ for d13CO2. We added this information to the manuscript.

L130-131. What about for the d13CO2? It is weird that author fixed they required optical path length based on the CH4 and N2O required precision rather than on the d13CO2 measurement.

Fig.2 shows the absorption of the different species with their corresponding axes (left and right). Please note the scaling difference between these axes. The CH4 and N2O absorptions are less than 0.5%, while those for  $CO_2$  are about 60 %. This clearly shows that the required optical path length was mainly driven by CH4 and N2O. We changed the wording to include this detail.

**L139. Significantly longer. Could you provide a value?!**

Up to several hours. The required duration was strongly dependent on the conditioning history of the cell, but even by baking, purging, and evacuating the cell over few days the pump-down time of the cell after filling with a sample gas took more than 20 minutes. This information has been added to the manuscript.

L155-156: How authors can simulate the frequency of the optical fringes produced by the multiple passages?! It is just a maximization of the distance between the neighboring spots as mentioned few lines below or is something else?

Yes, but not only. Besides the lowest possible overlap of the individual reflection spots, we also take into account the optical path difference of the neighboring spots and try to avoid those patterns where the interference fringe frequency generated by the optical path difference is comparable to the width of the absorption line. Especially, overlaps of the beam spots with small pass number (e.g. 4 and 6) differences are problematic. Searching for patterns that mostly fulfil these criteria is expected to result in a reduced optical fringe level or at least have less impact on the absorption line retrieval. These considerations have been added to the manuscript.

L203. Critical orifice. Can authors mention the size? Well, in flow measurement this would probably fractionate...please explain.

Added a short text to clarify this issue: "The critical orifice size (20  $\mu$ m) as well as the output pressure of the regulators were all carefully selected to avoid isotopic fractionation as the gases flow through the system. A series of experiments was conducted in flow-through and static regime to investigate potential fractionation effects due to flow-restriction elements such as critical orifice and the optional metering valve

**used in the flow-through mode. Their effect was found to be lower than our detection limit. With this setup it is possible to operate the instrument both in flow through or in batch mode for discrete samples."**

Table 1. When second digit precision is reported (eg. 0.01) then concentration should be reported with the same number of digits (eg. 248.8x ppm). Did authors monitor their standard mixtures over time? Do they see any drift?

Added the second significant digit to the values.

Figure 7. It would be interesting to see the same AW statistical analysis for a static experiment. Can authors add this information? Since the ice core measurements will certainly done using discrete samples and in static mode, I guess.

We performed several experiments of this kind, also because we had the same concern as the Referee. These tests resulted in very similar precision values as those conducted in flow-through mode. Of course, in this case we had to limit the measurement period to shorter time (< 1 h), to avoid drift effects caused by surface effects. The performance in batch mode is also demonstrated by the following sections of the manuscript, where we systematically focused our attention on discrete samples and demonstrated analytical performance obtained using such 1 mL samples. In our opinion, these data are more informative then showing a replicate of an Allan-Werle plot.

L281: This sequence is repeated several times: Can authors provide information about the frequency of those 50 measurements?

The time required to measure consecutive sample/reference pairs was 10 minutes, thus, for the 50 repetition cycle, 8.3 hours was necessary. We added this information to the revised manuscript.

L293-294. "Even for discrete samples...". What do they mean with this sentence? Because for a single discrete measurement the performances are maintained for a time given by the AW analysis, which is shown only a maximum integration time of 16 minutes. What authors mean with this 24h of continuous operation? Can they prove it with an AW plot?

Individual or discrete gas samples can be measured with a precision as determined by the AW analysis. However, when switching between sample gases collected in dip-tubes further additional artifacts/biases, mainly originating from gas handling, can affect the accuracy of the measurements. Of course, during the time that is required to replace the sample gas with the next sample the spectrometer can drift. However, we demonstrate that with our cyclic measurement approach, i.e. quickly alternating between dip-tube sample and reference gas measurement, these drifts can be accounted for and thus their effect can be minimized even for longer periods of time, e.g. 24 h. Figure 8 demonstrates exactly this situation.

L311-312: "This two-point calibration..." This is normally not a linear relationship, but rather an exponential with discrepancy becoming larger and larger at low concentrations...It is possible that authors would be able to approximate it to a linear relation in the range of concentrations they will have in the ice cores? Why authors did not just apply different dilution factors to on of their standard to study this effect on a larger concentration range? Just to mentioned, I see an artefact of 0.94 permil on the working range of 157.7 - 345.5 ppm, which is more the 20 times larger than the claimed accuracy.

In our experience, the linear relationship is a very good approximation (see also the fit residuals on Fig.9) for such a narrow concentration range. We did several measurements in the past using the dilution approach as also suggested by the Referee, and we always found a tight linear correlation even for various molecular species, e.g. Tuzson et al., Appl. Phys. B 92, 451–458 (2008) and Waechter et al, Opt. Exp. 16, 9239-9244 (2008). Our focus was only for the concentration range expected for ice core samples, because this is the main application for which we developed the instrument and which we fully cover with our custom-made standard gases. The dilution is feasible, but it is a time-consuming process that

we want to avoid when analyzing ice core samples. That's why we opt for a robust two-point verification approach. Unfortunately, we are unable to follow the Referee regarding the artefact.

Figure 9. So, if I understand well, the difference between the measured and the reference corresponds to the difference between the measurement done by the commercial instruments used for calibrate the standard gas mixtures and your new instrument, is that correct? But, you should say what is your approach for your instrument. Do you stick with the intensities provided by HITRAN database or you calibrate the spectrometer with one of the standard bottles? Because if it is the second option, then you should have at least one point which would match well the reference measurement. And what about the d13CO2? Because for that you have to rely on a reference measurement. So, I do not get why there is this offset between reference and measurement for all isotopic ratios rather than a crossing point.

Yes, that is correct. The concentration of the trace gases in the reference cylinders were determined using the WCC instrumentation, while the  $\delta^{13}C(CO_2)$  values were determined by IRMS, as described at L240-L243. The raw values reported by our spectrometer are purely based on spectroscopic (HITRAN) and physical parameters (OPL, T, and P) without any additional calibration. In the case of the  $\delta^{13}C(CO_2)$  the same approach was used, but included the scale conversion from natural abundance used by HITRAN and the *Rs* value defined by the VPDB-scale. Finally, a constant offset of 9 ‰ was added to bring the "spectroscopic"-scale closer to the VPDB values. This last step could also be included in the calibration function. The calibration curves in Figure 9 then illustrate the conversion of the instrument values to international scales. We revised the corresponding paragraph to include these details.

L343. "This indicates that..." You should explain your approach here. I believe that you leave the Lorentzian contribution on the absorption lines free to adjust with pressure. This should be mentioned in the text. As well as that the concentration in mixing ratio is then corrected for the total gas pressure in the cell. I believe that the artefact you see could be removed (or at least minimized) by changing the fit profile as mentioned above.

Exactly, the Lorentzian width is calculated based on the actual pressure readings and using the pressure broadening parameters  $\gamma_{air}$  while the concentration is determined using the ideal gas law. We added this information to the manuscript. We also think that the artifact shown in Fig. 10 could be reduced by using a more appropriate line profile model that includes at least the Dicke narrowing term. Further contributions can be due to the uncertainties in the pressure broadening coefficients, line strengths and frequency scaling as well as optical saturation.

L345. Can authors estimate the dependency for d13CO2 around 5 mbar? what precision in pressure is required the 0.04 permil precision and how it compares with your accuracy on the pressure measurement?!

Yes, the dependency of  $\delta^{13}C(CO_2)$  on pressure around 5 mbar is 0.81‰/mbar, thus we need 0.05 mbar accuracy on the pressure measurement to achieve 0.04 permil precision on  $\delta^{13}C(CO_2)$ . Our pressure sensor has a stated accuracy of 0.2% of the measured value, i.e., an uncertainty of 0.01 mbar for the 5 mbar sample pressure. We included this information in the revised manuscript.

L356: "possibly at value > = 5 mbar." Here you should rather report 5 +/- xx mbar, with the error estimated in order to stay within the 0.04 permil (comment above). Done.

L370-371: mention that those are 1 sigma precisions. Done.

Technical corrections:

L9. I would rather call it multi-pass absorption rather than direct absorption (or directmulti-pass absorption if authors want to keep the word direct).

We do not agree. The technique used here is the direct absorption spectroscopy, while the multipass is just to enhance the signal.

L61-80. This part should go to the introduction rather than in the methods. The latter should be focus on the description of the methods.

We agree and implemented the suggested changes.

L108. Mention the photodiode model. Done.

L173. I suggest By applying. Done.

L293: replace "targets" with "target performances" Done.

**Interactive comment by Thomas Bauska**

I found the paper very interesting to read throughout but had one question constantly in the back of my mind: how will the instrument deal with the presence of ice core drilling fluid in the gas sample? Of course, we do everything we can to avoid this and there are ways to mitigate the problem (that are beyond the scope of this study). I was thinking more along the lines of whether this was considered during the design phase. Or if the author's have been able to do theoretical or experimental work on potential spectral interferences. That's all. If drilling fluid can be shown not to be a problem, than the method will have (another) major advantage on dual-inlet IRMS measurements of carbon isotopes as drilling fluid can be the Achilles' Heel of this technique. However, if drilling fluid is problematic, it's not clear to me if traditional GC-MS techniques remain better suited for small samples.

Yes, we are well aware of this highly challenging issue and therefore we clarified beforehand the expected type of the drilling fluid that will be used within the "Beyond EPICA Oldest Ice Core" project. We will add the following lines to the manuscript:

"A significant advantage of laser spectrometry over - for instance - mass spectrometry is that the absorption lines are gas-specific and interferences between different gases unlikely. In the case of ice core analyses, in particular, the use of organic drill fluids may lead to contamination with potentially absorbing trace gases. Therefore, we tested for interferences with the drill fluid (ESTISOL™ 140) which is to be used in Antarctica within the Beyond EPICA ice core project. We purchased pure ESTISOL™ 140 and introduced the headspace of drill fluid contaminated sample into the multipass cell in quantities that we considered representative for the ice core samples. This lead to no alterations in the spectrum within the wavenumber window covered by our two lasers. The main reason is that these large organic compounds have very broad absorption features and tend to be localized to frequencies typical for functional groups. Therefore, our narrow-band laser spectroscopic approach in the mid-IR is not affected by such contaminations in the studied wavenumber range."

**High precision High-precision laser spectrometer for multiple greenhouse gas analysis in 1 mL air from ice core samples**

Bernhard Bereiter1,3, Béla Tuzson1, Philipp Scheidegger1,2, André Kupferschmid2, Herbert Looser1, Lars Mächler3, Daniel Baggenstos3, Jochen Schmitt3, Hubertus Fischer3, and Lukas Emmenegger1 1Laboratory for Air Pollution / Environmental Technology, Empa - Swiss Federal Laboratory for Materials Science and Technology, 8600 Dübendorf, Switzerland 2Transport at Nanoscale Interfaces, Empa - Swiss Federal Laboratory for Materials Science and Technology, 8600 Dübendorf, Switzerland 3Climate and Environmental Physics and Oeschger Center for Climate Research, University of Bern, 3012 Bern, Switzerland **Correspondence:** Béla Tuzson (bela.tuzson@empa.ch)

Abstract. The record of past global background atmospheric greenhouse gas composition from ice cores is crucial for our understanding of global climate change. The "Beyond EPICA Oldest Ice Core" project is currently pushing the frontier of this knowledge forward by the retrieval of an ice core reaching back to Future ice core projects will aim to extend both the temporal coverage (extending the time scale to 1.5million years ago. The oldest section of this core will have been strongly thinned by

5 glacier flow with about 15 kyr being trapped in as little as 1 m thickness of ice. This reduces the available sample volume to only a few mL of air for the targeted century-scale resolution of greenhouse gas records. Under these conditions, the required accuracy for multiple greenhouse gases cannot be achieved with currently available analytical methods.

Here, we present a new approach to unlocking such challenging atmospheric archives with a Myr) and the temporal resolution of existing records. This implies a strongly limited sample availability, increasing demands on analytical accuracy

- 10 and precision, and the need to reuse air samples extracted from ice cores for multiple gas analyses. To meet these requirements, we designed and developed a new analytical system that combines direct absorption laser spectroscopy in the mid-infrared with a quantitative sublimation extraction method. Here, we focus on the high-precision mid-IR-dual-laser direct absorption spectrometer. The instrument is designed to simultaneously measure spectrometer, for the simultaneous measurement of  $CH_4$ ,  $N_2O$ , and  $CO_2$  concentrations, as well as  $\delta^{13}C(CO_2)$  using. Flow-through experiments at 5 mbar gas pressure demonstrate an
- 15 analytical precision  $(1 \sigma)$  of 0.006 ppm for CO2, 0.02% for  $\delta^{13}C(CO_2)$ , 0.4 ppb for CH4 and 0.1 ppb for N2O, obtained after an integration time of 100 s. Sample-standard repeatabilities  $(1 \sigma)$  of discrete samples of only 1 mL STP-1 ml STP amount to 0.03 ppm, 2.2 ppb, 1 ppb and 0.04% for CO2, CH4, N2O and for  $\delta^{13}C(CO_2)$ , and it achieves a precision of 1.6 ppb, 1.0 ppb, 0.03 ppm and 0.04%, respectively. Repeated 
[revised manuscript text omitted]
                                                 | ∐D | $\overset{\nu}{\sim}$ | $\stackrel{S}{\sim}$ | $E''_{\sim}$ | $\stackrel{A}{\sim}$ |
|---------------------------------------------------------|-----------|-----------------------|----------------------|--------------|----------------------|
| $\overbrace{\sim}^{12} \underbrace{\text{CO}_2}_{\sim}$ | 21        | 2301.680904           | 2.71                 | 1276.4476    | 202.3                |
| $\overset{13}{\sim} \overset{\text{CO}_2}{\sim}$        | 22 | 2302.308939           | 2.62                 | 273.8809     | 187.8                |
| $N_2O$                                                  | 41        | 1301.684840           | 15.40                | 175.9536     | .6.1          |
| $\mathrm{CH}_4$                                         | 61 | 1302.044313           | 6.45          | 219.9199     | 2.2                  |

[revised manuscript text omitted]
 100 MHz). The first condition is well accomplished met by a base length of 20 cm and mirror